# Identification of plasma proteomic markers underlying polygenic risk of type 2 diabetes and related comorbidities

Douglas P. Loesch [1] ✉, Manik Garg [1], Dorota Matelska[1], Dimitrios Vitsios [1], Xiao Jiang[1], Scott C. Ritchie [2,3,4,5,6], Benjamin B. Sun [7], Heiko Runz [7], Christopher D. Whelan [8], Rury R. Holman [9], Robert J. Mentz[10], Filipe A. Moura[11,12,13], Stephen D. Wiviott[11], Marc S. Sabatine[11], Miriam S. Udler [14,15,16,17], Ingrid A. Gause-Nilsson[18], Slavé Petrovski [1], Jan Oscarsson [18], Abhishek Nag[1,21], Dirk S. Paul [1,19,21] ✉ & Michael Inouye[2,3,4,5,6,20,21]

Genomics can provide insight into the etiology of type 2 diabetes and its comorbidities, but assigning functionality to non-coding variants remains challenging. Polygenic scores, which aggregate variant effects, can uncover mechanisms when paired with molecular data. Here, we test polygenic scores for type 2 diabetes and cardiometabolic comorbidities for associations with 2,922 circulating proteins in the UK Biobank. The genome-wide type 2 diabetes polygenic score associates with 617 proteins, of which 75% also associate with another cardiometabolic score. Partitioned type 2 diabetes scores, which capture distinct disease biology, associate with 342 proteins (20% unique). In this work, we identify key pathways (e.g., complement cascade), potential therapeutic targets (e.g., FAM3D in type 2 diabetes), and biomarkers of diabetic comorbidities (e.g., EFEMP1 and IGFBP2) through causal inference, pathway enrichment, and Cox regression of clinical trial outcomes. Our results are available via an interactive portal (https://public.cgr.astrazeneca.com/t2d-pgs/v1/).

Diabetes mellitus is a complex, multifactorial metabolic disorder diagnosed via a single clinical feature, hyperglycaemia[1,2]. Among the major diagnoses of diabetes mellitus, type 2 diabetes (T2D) has the largest worldwide disease burden[1]. To date, large-scale genome-wide association studies (GWAS) for T2D[3,4] have helped uncover important T2D biology, such as the link between *C2CD4A* and beta-cell dysfunction[5,6]. However, many T2D risk variants identified via GWAS have small effect sizes and tend to map to non-protein-coding regions, which makes it challenging to uncover their downstream biological effects. Polygenic risk scores (PRS), also known as polygenic scores (PGS), aggregate the small effects of these variants and can be utilised for risk prediction and stratification, including for T2D[7,8].

Pairing PGS with molecular data can provide a powerful approach to understanding the downstream biological effects of polygenic risk to disease. This can help elucidate key pathways relevant to disease pathophysiology and potentially find therapeutic targets that may be missed by traditional disease-gene mapping approaches. For example, studies by Ritchie et al. and Steffen et al. tested the association between PGS and protein expression levels, uncovering molecular mechanisms underlying polygenic risk of cardiometabolic diseases[9,10]. Furthermore, PGS and proteomics information, coupled with causal inference methodology such as mediation, complements approaches that use individual genetic variants as instruments, such as Mendelian randomisation (MR). This

is particularly crucial for proteins that lack the genetic variation (i.e., large enough effect size and/or frequency) needed to serve as MR instruments.

T2D is highly heterogeneous, with affected individuals having different degrees of insulin resistance and beta-cell dysfunction[11]. In addition, the vast majority of T2D patients have at least one additional comorbidity, such as hypertension, obesity or hyperlipidaemia[12], resulting from the complex interplay between the pathophysiology of T2D, adiposity, and the other comorbidities. A single T2D PGS is unlikely to capture this heterogeneity. To address this, partitioned polygenic scores (pPS), derived from the genetic clustering of GWAS-identified T2D variants, have been developed to capture biological processes underlying T2D genetic risk[13,14]. These pPS improve the prediction of clinical outcomes in patients with T2D compared to a conventional genome-wide T2D PGS[15]. Based on these observations, we hypothesise that leveraging genome-wide and partitioned T2D PGS, cardiometabolic PGS, and large-scale proteomics data will identify proteins and pathways that lead to the development of T2D or comorbidity and discover molecular mechanisms where these processes intersect.

In this work, we interrogate how PGS for T2D (including five partitioned T2D scores[13]: beta cell, lipodystrophy, liver lipid, obesity, proinsulin) and its cardiometabolic complications (coronary artery disease - CAD, chronic kidney disease - CKD and adiposity ascertained as body mass index - BMI) perturb the plasma proteome using data from the UK Biobank Pharma Proteomics Project (UKB-PPP)[16,17]. We further perform causal inference in the UK Biobank and survival analysis for cardiorenal outcomes in randomised controlled trials (RCTs) using the PGS-associated proteins to identify potential therapeutic targets and biomarkers of T2D comorbidities. By leveraging a large-scale biobank and two RCTs, our study provides insights into the etiology of T2D and its comorbidities that may have translational potential.

## Results

### Study design
This study leverages proteogenomic data from three studies: the UK Biobank[18] (UKB), Exenatide Study of Cardiovascular Event Lowering[19] (EXSCEL), and Dapagliflozin Effect on Cardiovascular Events–Thrombolysis in Myocardial Infarction 58[20] (DECLARE–TIMI 58). The UKB is a population-level biobank with 14 years of follow-up time, with proteomic data available as part of the UK Biobank Pharma Proteomics Project (UKB-PPP). EXSCEL and DECLARE-TIMI 58 were both cardiovascular outcome trials in patients with T2D with mean follow-up times of 3.2 and 4.2 years, respectively (see **Methods** for a full cohort description). We first tested T2D and cardiometabolic PGS for association with circulating proteins in the UKB, followed by further analyses to identify putatively causative proteins among the set of PGS-associated protein biomarkers. Then, to assess the association between PGS-associated proteins and common comorbidities, we used cardiorenal outcomes in EXSCEL and DECLARE-TIMI 58. See Fig. 1 for an overview of our workflow, **Methods**, and Supplementary Data 4 and 5 for descriptions of all PGS.

### Associations of T2D PGS with protein expression levels
To determine how polygenic risk for T2D impacts the circulating proteome, we tested PGS for T2D, including the partitioned scores, for association with circulating proteins in the UKB-PPP. The genome-wide T2D PGS ($PGS_{T2D\_gw}$) was associated with 648 proteins in the UKB discovery set (Table 1); of these, 617 replicated in the UKB replication set (FDR < 5%). The proteins that were among the top 1% in terms of variance ($R^2$) explained by the $PGS_{T2D\_gw}$ include PON3, CKB, APOF, and IGFBP2 (Fig. 2A). The partitioned T2D scores and the $PGS_{T2D\_gwas}$ (i.e., the PGS derived from GWAS-significant variants) were associated with fewer proteins compared to the genome-wide T2D score

($PGS_{T2D\_gw}$) (Supplementary Figs. 2–6 and Supplementary Data 6). Despite comprising fewer variants than $PGS_{T2D\_gw}$, three of the partitioned T2D scores were significantly associated with proteins that were not associated with the $PGS_{T2D\_gw}$: $PGS_{T2D\_beta\_cell}$ (76% of its associations), $PGS_{T2D\_liver\_lipid}$ (55%), and $PGS_{T2D\_lipodystrophy}$ (8%). When comparing the beta coefficients of the different T2D PGS for the circulating proteins, we found that they were negatively correlated between $PGS_{T2D\_gw}$ and $PGS_{T2D\_liver\_lipid}$ (Pearson's $r = -0.19$, $p = 5.9 \times 10^{-25}$) and between $PGS_{T2D\_gw}$ and $PGS_{T2D\_proinsulin}$ (Pearson's $r = -0.07$, $p = 3.8 \times 10^{-4}$, Fig. 2C). While their effects on the circulating proteome differ, the actual $PGS_{T2D\_gw}$ and $PGS_{T2D\_liver\_lipid}$ scores are positively correlated (see Supplementary Fig. 7). Overall, this suggests that the partitioned T2D scores capture protein associations representing perturbations in specific T2D-related biological pathways that may be obscured when variant effects are aggregated in a genome-wide score ($PGS_{T2D\_gw}$).

### Impact of adiposity on T2D score-protein associations
T2D and adiposity are highly interconnected[21], and as such, protein-T2D score associations can be confounded or mediated by adiposity. To explore which protein associations are independent of adiposity, we added one of three anthropometric measures capturing obesity as covariates to our PGS-protein models: body mass index (BMI), waist circumference (WC), and waist-hip ratio (WHR). Out of the 648 proteins significantly associated with the $PGS_{T2D\_gw}$ in the discovery set, 314 remained significant after adjustment for BMI (Fig. 2B), demonstrating the close interplay between T2D genetic risk, BMI, and circulating protein levels. Adjusting for WC or WHR instead of BMI produced similar results (see Supplementary Fig. 8 and Supplementary Data 7), with associations for 301 and 360 proteins remaining significant, respectively.

Next, to explore the extent to which adiposity is the mechanism by which the T2D scores exert their influence on the plasma proteome, we performed a mediation analysis with the T2D scores as the exposure, BMI as the mediator, and protein expression levels as the outcome (see Supplementary Data 7, **Methods**). For 94 of the 617 proteins significantly associated with the $PGS_{T2D\_gw}$, the indirect effect and total effects were significant ($p$-value $< 4.3 \times 10^{-6}$), but the direct effect of the PGS was not. This suggests that for these proteins, such as ADM, LEP and TNF, BMI mediated the bulk of the $PGS_{T2D\_gw}$'s effect on the circulating levels. For 518 proteins, the direct effect of $PGS_{T2D\_gw}$ remained significant, but BMI still mediated part of the $PGS_{T2D\_gw}$'s effect on the proteins: the proportion mediated by BMI ranged from relatively small (0.05 for MANSC1) to relatively large (0.82 for FABP4), with a median of 0.38 (see Supplementary Fig. 8C). For 12 proteins, the indirect and direct effects were in the opposite direction. One of these proteins, TIMP4, has been previously reported to have a discordant relationship with adiposity and T2D[21]. Among the partitioned T2D scores, it was only $PGS_{T2D\_obesity}$ that the bulk of the effect was mediated via BMI.

### Cardiometabolic score associations
We then tested a selection of cardiometabolic PGS representing common T2D comorbidities (CAD, CKD, and BMI) for their association with protein expression levels (Panel A from Supplementary Figs. 9–11). This analysis aimed to assess proteomic signatures that are specific to and shared across polygenic risk for T2D and its comorbidities. The proteins associated with the $PGS_{T2D\_gw}$ were frequently associated with one or more cardiometabolic PGS (see Fig. 2D). Overall, 64% were also significantly associated with the $PGS_{BMI}$, 33% were associated with the $PGS_{CKD}$, and 17% were also associated with the $PGS_{CAD}$. The beta coefficients of the various comorbidity scores on the circulating protein levels were all positively correlated with that of the $PGS_{T2D\_gw}$ ($p$-value $< 0.05$, Fig. 2C), supporting the epidemiological observation that cardiometabolic diseases are highly interconnected[22,23].

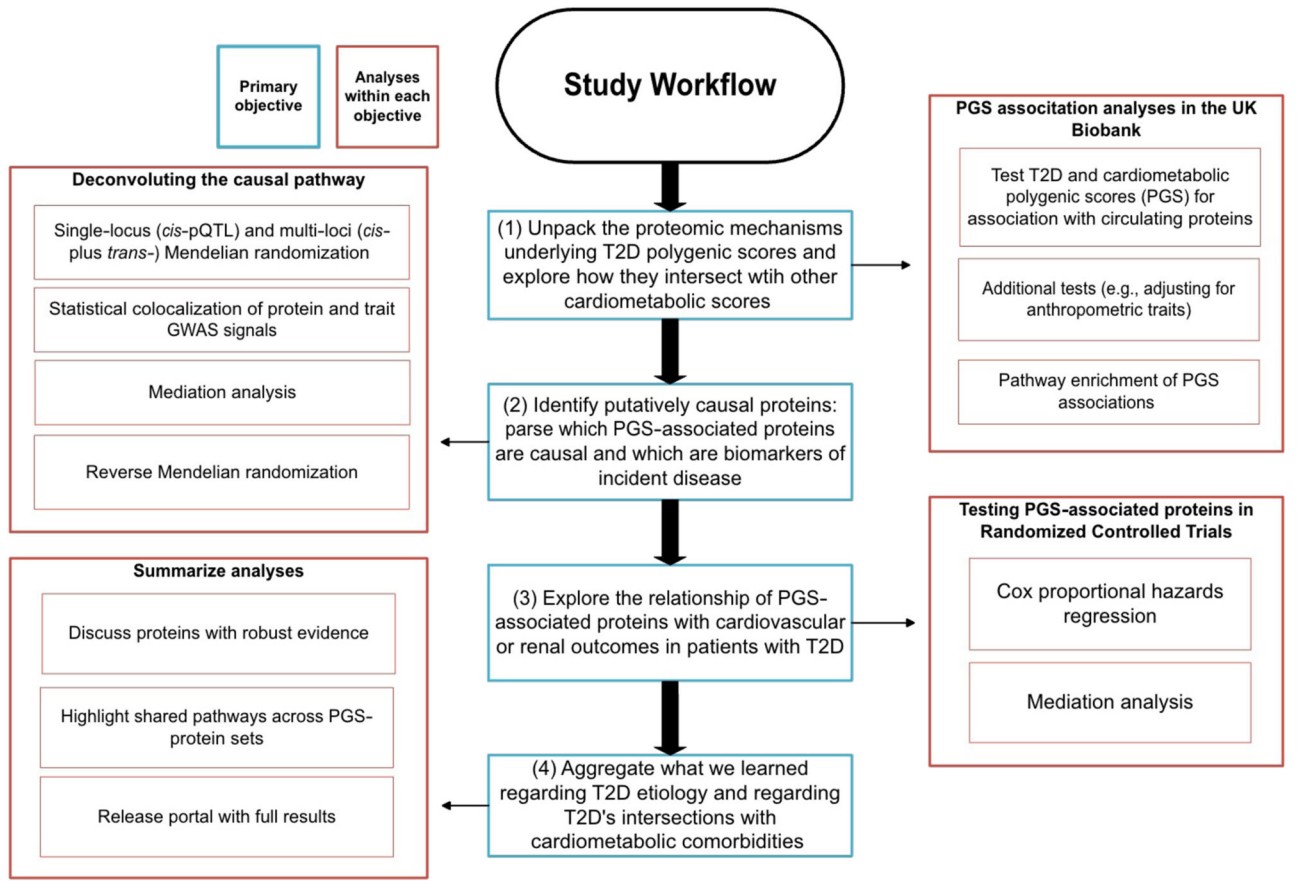

**Fig. 1 | Study workflow.** The centre of the flowchart corresponds to the primary objectives or aims of the study (in **Blue**). Branching off each objective are short summaries of the analyses corresponding to that objective (in **Red**).

## Table 1 | Summary of polygenic score (PGS) associations

| PGS Category | PGS | Top protein assoc. | N significant proteins | N significant after pQTL/BMI adjustment | % Unique assoc. |
|---|---|---|---|---|---|
| T2D (Genome-wide) | $PGS_{T2D\_gw}$ | IGSF9 (+) | 617 (648) | 381 (310) | 18 |
| T2D (GWAS-significant only) | $PGS_{T2D\_gwas}$ | PAM (−) | 61 (88) | 51 (61) | 2 |
| Partitioned T2D polygenic scores | $PGS_{T2D\_beta\_cell}$ | ABO (+) | 21 (30) | 6 (4) | 29 |
| | $PGS_{T2D\_lipodystrophy}$ | CD300LG (−) | 97 (137) | 96 (119) | 4 |
| | $PGS_{T2D\_liver\_lipid}$ | LEPR (−) | 306 (392) | 13 (9) | 22 |
| | $PGS_{T2D\_obesity}$ | LEP (+) | 3 (15) | 2 (11) | 0 |
| | $PGS_{T2D\_proinsulin}$ | FOLR3 (−) | 0 (0) | 0 (0) | − |
| Other cardiometabolic scores | $PGS_{CAD}$ | GRN (+) | 130 (159) | 102 (78) | 6 |
| | $PGS_{CKD}$ | HLA-A (+) | 556 (584) | 534 (515) | 29 |
| | $PGS_{BMI}$ | LEP (+) | 626 (694) | 625 (662) | 11 |

PGS Category: descriptive information for the PGS. PGS: Name of tested polygenic score. Top protein association: protein with the lowest *p*-value in the combined UKB-PPP cohort (discovery + replication) for each PGS. N significant proteins: number of significant proteins after replication. The brackets specify the total number of proteins that were significant in the discovery subset. N significant after pQTL/BMI adjustment: number of replicated proteins remaining significant after pQTL (protein quantitative trait loci) and BMI (body mass index) adjustments. The brackets specify the total number of proteins that were significant after adjustment in the discovery subset. For the $PGS_{BMI}$ and the $PGS_{T2D\_obesity}$, we did not also adjust for BMI. % Unique assoc.: percentage of significant PGS-associated proteins that were only associated with that PGS.

However, for a subset of proteins, the directions of the effect of the $T2D_{T2D\_gw}$ and a comorbidity PGS on circulating levels were opposite (e.g., the effect of $PGS_{T2D\_gw}$ and $PGS_{CKD}$ on MANSC4 levels, see Supplementary Data 8), possibly indicating more complicated relationships such as compensatory responses[21,24].

### Genetic ancestry and PGS-protein associations
To examine whether our findings are robust and portable across ancestries, we compared and contrasted PGS-protein association patterns across five genetically predicted ancestry groups from the UKB-

PPP cohort (see **Methods**). As over 92% of the UKB-PPP cohort is of European ancestry, we compared beta coefficients (rather than *p*-values) of the PGS-protein associations between the different ancestry groups. The correlations of the PGS-protein beta coefficients between genetically predicted European and non-European ancestries were less than 1 (Pearson's r), reflecting the fact that translatability of PGS and, by extension, their proteomic associations, across ancestries is a potential concern (see Supplementary Figs. 12–15 and Supplementary Data 6). However, the beta coefficients were notably more correlated across the ancestry groups when we restricted our

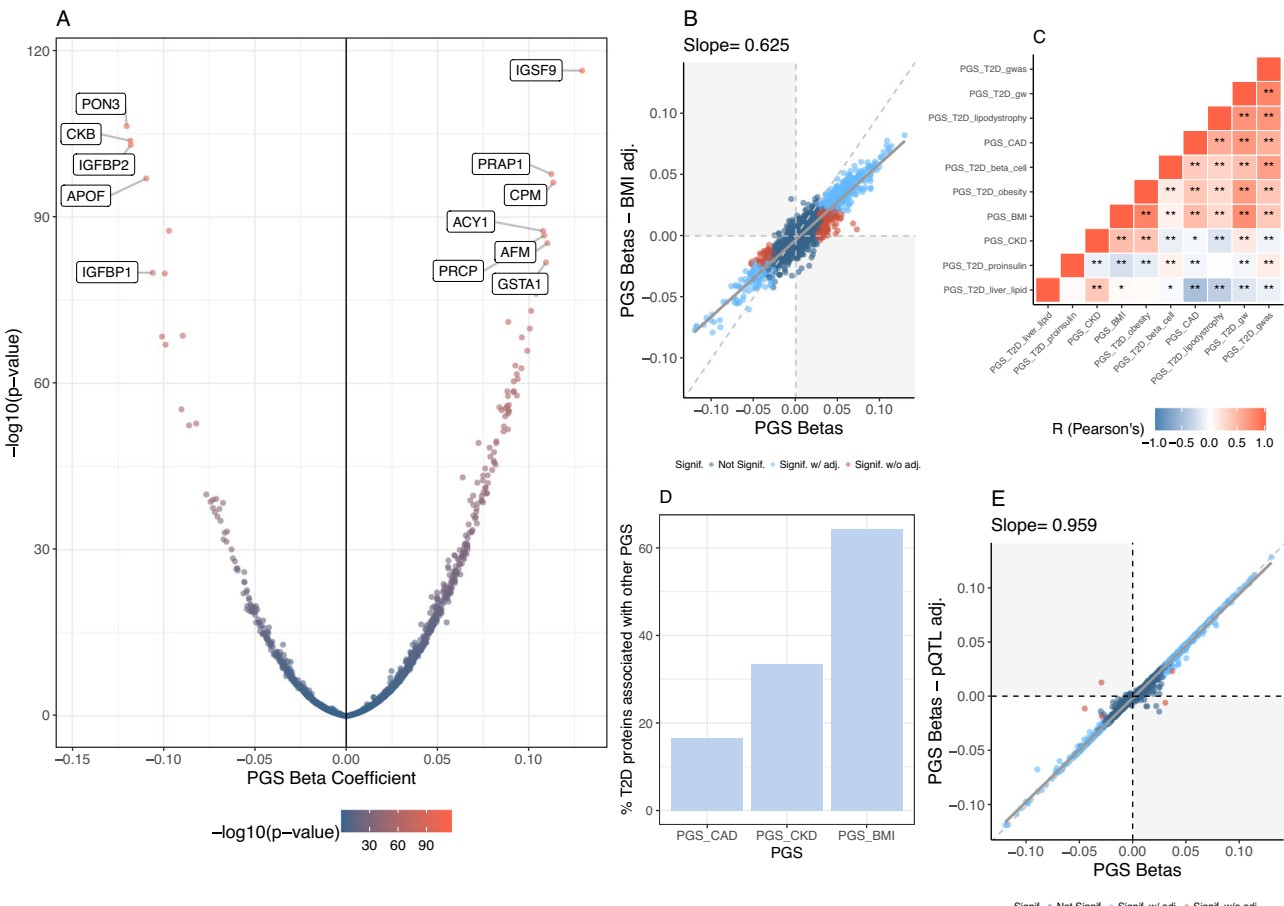

**Fig. 2 | Polygenic score (PGS) associations with circulating proteins. A** Volcano plot of PGS$_{T2D\_gw}$-protein beta coefficients (obtained from linear regression) and the unadjusted -log$_{10}$ $p$-values (two-sided), with the colour indicating the magnitude of the -log10 $p$-values. Labelled proteins are among the top 1% in terms of variance (R$^2$) explained by the PGS$_{T2D\_gw}$. **B** Beta-beta plot of PGS$_{T2D\_gw}$ beta coefficients on circulating proteins with ($y$-axis) and without ($x$-axis) BMI adjustment. The diagonal is dashed grey, while the regression line is solid grey. Each point represents a protein; light blue points indicate replicated proteins that remained significant with the adjustment, red points indicate replicated proteins that were no longer significant after the adjustment, and dark blue points indicate proteins that did not significantly replicate prior to adjusting for BMI or pQTLs. **C** Pearson's correlations of PGS beta coefficients from the regression on circulating protein

levels. Red indicates pairs of PGS with positively correlated effect sizes; blue indicates negatively correlated effect sizes. "*" indicates correlations with a $p$-value < 0.05 and "**" indicates correlations with a $p$-value < 0.001 (a Bonferroni correction for 45 comparisons). $P$-values are unadjusted and two-sided $t$ test as the test statistic follows a $t$ distribution. **D** Bar plot indicating the overlap between proteins significantly associated with the T2D PGS and the other cardiometabolic PGS. The $x$-axis is the PGS label, and the $y$-axis is the percentage of PGS$_{T2D\_gw}$-associated proteins that are also associated with another PGS (e.g., over 60% of proteins were also associated with the PGS$_{BMI}$). **E** Beta-beta plot of PGS$_{T2D\_gw}$ effect sizes on circulating proteins with ($y$-axis) and without ($x$-axis) pQTL adjustment, with the same definitions as panel (**B**) albeit for a pQTL adjustment.

comparison to the statistically significant PGS-protein associations, suggesting that our replication strategy was effective and that significant associations are more likely to be robust and translatable across ancestries.

## Polygenicity of PGS-protein associations and widespread effects of GCKR

A key motivation of our PGS-based approach is that it can help detect proteomic associations that may otherwise be undetectable via single-variant analysis (i.e., protein quantitative trait loci or pQTL analysis). To evaluate whether the PGS-protein associations we detected were driven by a single pQTL tagged by the PGS or represent the cumulative, polygenic effect of variants in the PGS, we additionally tested the PGS-protein associations by including previously reported pQTLs as covariates in the regression model. After adjusting for both *cis* and *trans* pQTLs (see **Methods**), most protein-PGS$_{T2D}$ associations remained significant (612 out of 617; see Fig. 2E, Table 1, Supplementary Data 6), demonstrating that these associations were indeed polygenic in nature. This was largely true for all evaluated PGS. However, for the PGS$_{T2D\_liver\_lipid}$ and the

PGS$_{T2D\_beta\_cell}$, most protein associations were not significant after pQTL adjustment (Supplementary Fig. 2C and 4C). In the case of the PGS$_{T2D\_liver\_lipid}$, the index variant at the *GCKR* locus (rs1260326) explained the bulk of its association signature (280 out of 306 proteins), highlighting the pleiotropic effect of this variant on circulating proteins. In contrast, most proteins associated with the PGS$_{T2D\_beta\_cell}$ score were explained by 7 different pQTLs (four from the *ABO* locus, one from the *NCR3LG1* locus, and one from the *FGFBP3* locus).

## Causal inference using Mendelian randomisation and cis instruments

To assess if a PGS-protein association represents forward causality (i.e., the PGS perturbs a protein's expression level that leads to disease) and, therefore, characterise putatively causal proteins for T2D and its key comorbidities (BMI, CAD, and CKD), we utilised two-sample Mendelian randomisation (MR) with statistically independent pQTLs as instruments. We applied four conventional MR methods (IVW, median, weighted median, and MR-Egger; see **Methods**) to those proteins with sufficient genetic instruments.

Six proteins had significant causal evidence with regards to T2D (median *p*-value across the four MR methods <0.05, FDR-adjusted) without evidence of pleiotropy (MR-Egger intercept *p*-value > 0.05): MANSC4, GLRX5, NUCB2, PAM, ARG1, and NCR3LG1, (see Fig. 3A and Supplementary Data 9), with PAM and MANSC4 significant after a more stringent Bonferroni correction (*p*-value < $9.2 \times 10^{-6}$). For BMI, 14 proteins were significant using *cis* instruments (see Supplementary Fig. 16A), with NADK remaining significant after a Bonferroni correction (*p*-value < $8.7 \times 10^{-6}$). For CAD, we identified eight proteins using *cis* instruments (FDR-adjusted *p*-value < 0.05; see Fig. 3D), with FES, LPA and PCSK9 significant after a Bonferroni correction. For CKD, we identified one protein using *cis* instruments, UMOD (with UMOD significant after a Bonferroni correction; see Supplementary Fig. 16D). Finally, we conducted an MR analysis for cardiovascular and renal outcomes in UKB participants with T2D. Using *cis* instruments, we identified two proteins: UMOD and CCN4 (FDR *p*-value < 0.05; see Supplementary Fig. 17A).

We also employed a pleiotropy-robust method designed for *cis* molecular data, MR-Link-2[25] (see **Methods**). As MR with only one locus can be underpowered and can present challenges for pleiotropy estimation, this MR method complements our conventional MR analysis. MR-Link-2 and the conventional MR were largely concordant, though MR-Link-2 did identify an additional 12 proteins that were not found using the other MR methods (see Supplementary Data 10). Ultimately, using *cis* instruments, we found evidence of a causal link for 44 proteins, 70% of which were associated with at least one PGS.

**Mendelian randomisation with the addition of trans instruments**

After adding *trans* instruments in our MR models, we found 4 more proteins with significant causal evidence for T2D (LGALS4, MENT, PDIA5, and PPM1B, FDR-adjusted *p*-value < 0.05), with LGAL34 significant after Bonferroni correction (*p*-value < $8.7 \times 10^{-6}$). For BMI, we identified an additional 7 proteins (see Supplementary Fig. 16B), four of which were also significant after a Bonferroni correction (APOBR, PRSS53, GYS1, RAP1A). For CAD, six additional proteins were associated with CAD (FDR *p*-value < 0.05; see Fig. 3E), with FURIN and LMOD1 significant after a Bonferroni correction (*p*-value < $8.7 \times 10^{-6}$). No additional proteins were added after the inclusion of *trans* instruments for CKD. For the comorbidities in UK Biobank participants with T2D, we found that SELE increased the risk of myocardial infarction.

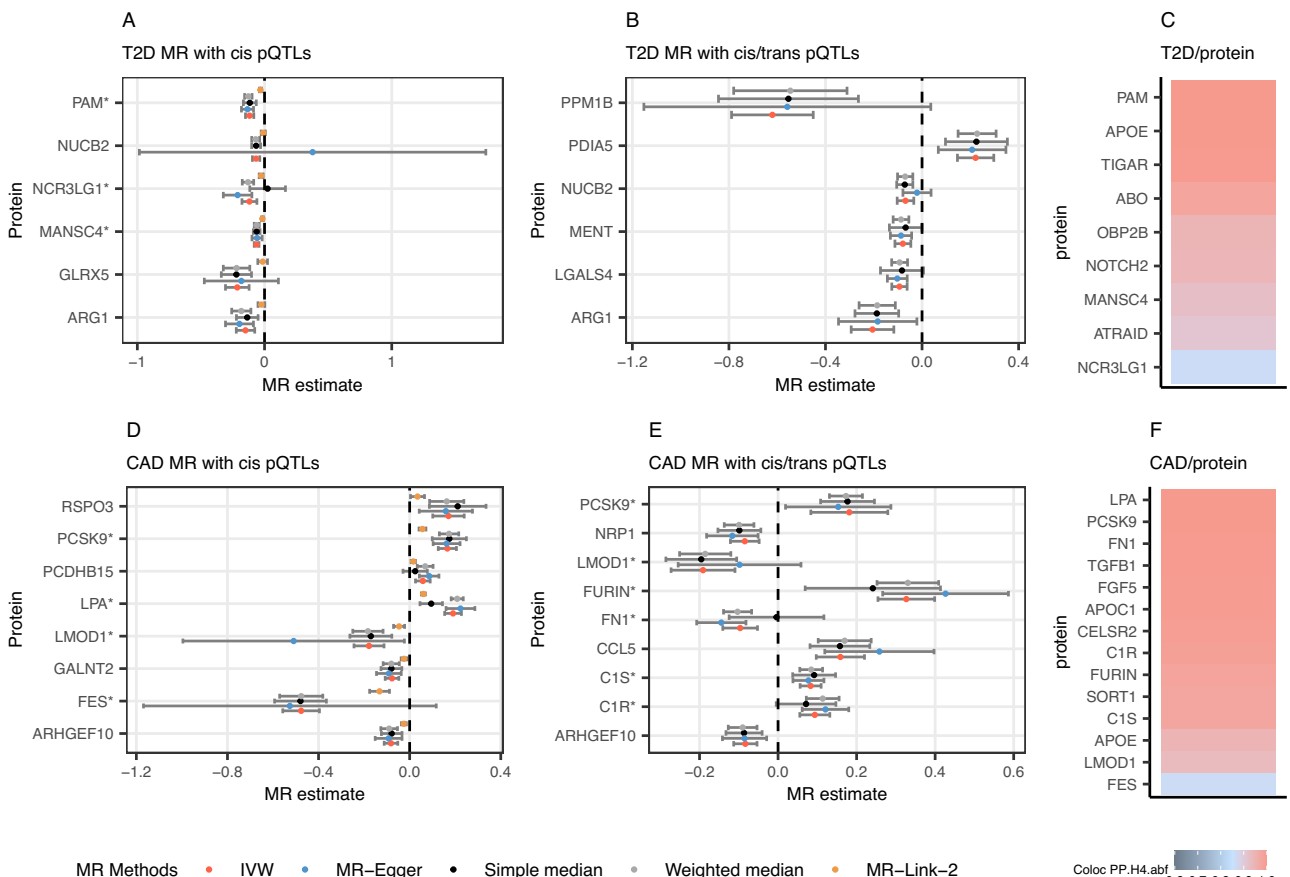

**Fig. 3 | Two-sample Mendelian randomisation analysis in the UK Biobank.**
**A** Type 2 diabetes (T2D) Mendelian randomisation (MR) with cis instruments for each protein as the exposure. **B** T2D MR with both cis and trans instruments for each protein as the exposure. **C** Cis colocalization using T2D and protein quantitative loci (pQTL) genome-wide association study (GWAS) information.
**D** Coronary artery disease (CAD) MR with cis instruments for each protein as the exposure. **E** CAD MR with both cis and trans instruments for each protein as the exposure. **F** CAD colocalization using CAD and pQTL GWAS information. In the MR plots, four conventional MR methods are displayed (simple median, weighted median, IVW, MR-Egger), plus an additional MR method called MR-Link-2. All proteins displayed in this figure had a median *p*-value across the four conventional MR methods < 0.05 (FDR-adjusted) and no pleiotropy as detected by MR-Egger (MR-Egger intercept *p*-value > 0.05). For panels (**A**, **B**, **D**, and **E**), the points represent the causal estimate obtained by each MR method (in the log-odds scale) and error bars represent the 95% confidence interval calculated using the standard error for each MR estimate. Note that MR-Link-2 estimates are on a different scale than the other MR methods but show consistency in the direction of effect. Finally, "*" signifies proteins with colocalization evidence. For panels A through F, the exposure (protein) GWAS had a sample size of 34,557 European-ancestry UK Biobank participants, and the outcome GWAS (T2D, CAD) had a sample size of 409,048 non-overlapping European-ancestry UK Biobank participants.

## Multivariable Mendelian randomisation with BMI and T2D

For BMI and T2D, due to their high intercorrelation (see above), we also employed multivariable MR (MVMR)[26]. By using genetic variants that impact both BMI and protein expression levels as MR instruments, we can estimate the effects of both exposures, even if they are related (i.e., through mediation). We identified 55 loci (cis regions of proteins) with sufficient instrument strength and without evidence of pleiotropy, to simultaneously evaluate both BMI and protein exposures on T2D (see Supplementary Data 11). Using this approach, we found 11 with nominal evidence (p-value < 0.05) that the BMI exposure (3 proteins), the protein exposure (7 proteins), or both (1 protein) lead to T2D risk, with one, FAM171B, significant after a multiple testing correction (FDR p-value < 0.05; see Supplementary Fig. 17B and C). Including trans instruments allowed us to evaluate 19 additional proteins with MVMR. Of these, 7 had nominal evidence (p-value < 0.05) that the BMI exposure (4 proteins, including TNF), the protein exposure (2 proteins), or both (1 protein) conferred T2D risk, with UROD significant after an FDR correction.

## Colocalization of T2D and comorbidity GWAS signals with UK Biobank pQTLs

We performed statistical colocalization to test whether pQTL and T2D/comorbidity GWAS signals arise from the same causal variant, thus providing additional support for our MR analyses. The cis regions of 9 proteins colocalized with T2D GWAS signals (see Fig. 3C), 31 proteins for BMI GWAS signals (Supplementary Fig. 16C), 14 proteins for CAD (Fig. 3F), and 3 for CKD (Supplementary Fig. 16E). For trans pQTLs, 70 had evidence for colocalization with T2D GWAS signals, corresponding to 502 proteins, suggesting that trans effects plays a large part in the genetic risk for complex disease as previously proposed[27] (see Supplementary Data 9). Similarly, 117 trans pQTLs had evidence for colocalization with BMI GWAS signals. For the comorbidities in UKB participants with T2D, one signal was colocalized (UMOD and CKD).

## The mediation of PGS effects by circulating proteins in the UKB

An orthogonal approach to MR for inferring the causal pathway is mediation, which tests whether a protein mediates the effect of a PGS on incident disease risk. Mediation can be used to support MR findings, and it can also provide information on directionality for proteins that lack the requisite number of genetic instruments for MR. Among the UKB participants with proteomics data, 2081 were diagnosed with T2D during 14 years of follow-up time. In our mediation analysis, we modelled the PGS as the exposure, an individual protein as the mediator, and either incident T2D, CKD, or CAD as the outcome. Ultimately, 520 of the 617 $PGS_{T2D\_gw}$-associated proteins significantly mediated the effect of the PGS on incident T2D risk. After adjusting for BMI, this is reduced to 238 proteins (Supplementary Fig. 18A and Supplementary Data 12). We also performed mediation using the partitioned scores (Supplementary Fig. 18B): 83 proteins mediated the effect of the $PGS_{T2D\_lipodystrophy}$ score (74 after BMI adjustment) and 6 mediated the effect of the $PGS_{T2D\_beta\_cell}$ score (4 after BMI adjustment). Finally, we also performed mediation analysis for the $PGS_{CKD}$ with incident CKD and the $PGS_{CAD}$ with incident CAD, finding 470 (422 after BMI adjustment) and 33 (12 after BMI adjustment) mediating proteins, respectively (Supplementary Fig. 18A).

## Reverse Mendelian randomisation with cardiometabolic traits

Next, we employed reverse MR to identify instances of reverse causation, i.e., where the cardiometabolic trait (T2D, CAD, CKD, BMI) alters the protein expression levels (**Methods**). If a PGS-associated protein is implicated in reverse MR, this suggests that the developing disease state affects the levels of the protein. After applying a Bonferroni correction as in our primary PGS-protein association analysis (p-value < $4.3 \times 10^{-6}$ across 4 MR methods), our reverse MR analysis suggested that circulating levels of 38 proteins were influenced by T2D, including GDF15 (Supplementary Data 13), and several other proteins strongly associated with the $PGS_{T2D\_gw}$ (e.g., APOF, PON3, PRCP). In the case of GDF15, elevated serum levels have been reported in T2D and it is known to play a role in regulating food intake and metabolism[28]. However, none of the proteins identified in the forward MR analysis for T2D were implicated in our reverse MR. It is worth noting that proteins identified via reverse MR could still be causal for other comorbidities or influence T2D risk via feedback mechanisms.

For the other cardiometabolic disorders, we found 3 proteins influenced by CAD risk (MMP12, CNTN4, LGALS4), 8 by CKD risk, and 539 by BMI (Supplementary Data 13), using reverse MR analysis. The high number of proteins influenced by BMI genetic risk indicates that the levels of many circulating proteins are impacted by adiposity levels. For CKD and CAD, none of the proteins we identified in our forward MR were associated in the reverse MR, though this was the case for 7 of the 21 BMI-associated proteins (CXCL16, LGALS3, GALNT10, PSCA, IL12RB2, ITGAL, SERPINA7). However, when we overlaid our mediation results with the reverse MR results, a proportion of mediating proteins had evidence for reverse causality, ranging from 2% of proteins for CKD and $PGS_{CKD}$ to 7% for the $PGS_{T2D\_gw}$.

## Time-to-event analyses of PGS-associated proteins in EXSCEL and DECLARE

As PGS-associated proteins could have clinical relevance as biomarkers for diabetic comorbidities and, in some cases, could be causally linked, we tested their association with cardiovascular and renal clinical trial endpoints in the placebo arms of EXSCEL and DECLARE-TIMI 58 (DECLARE).

In EXSCEL, the baseline levels of 241 proteins were significantly associated with time to a major adverse cardiovascular event (MACE), time to hospitalisation for heart failure (HHF), or time to the renal outcome (see **Methods**). Of these 241 proteins, 50% were significantly associated with the $PGS_{T2D\_gw}$, 72% were significantly associated with the $PGS_{CKD}$, and 9% were significantly associated with the $PGS_{CAD}$ in the UKB PGS analysis. After adjusting for clinical risk factors, 118 proteins remained significant (see Fig. 4A and Supplementary Data 14). Notably, 3 proteins were also independent of NT-proBNP for time to HHF (MARCO, EGFR, and EFEMP1), as well as 20 proteins for MACE. In DECLARE, 64 proteins associated with an outcome in EXSCEL were available for replication, of which 20 were significantly replicated using the corresponding outcome in DECLARE (Fig. 4B and Supplementary Data 15). When adjusting for clinical risk factors, 6 proteins were significantly associated with either MACE or HHF (FDR-adjusted p-value < 0.05), while for the time to DECLARE's composite renal outcome, IGFBP6 was nominally significant (p-value = 0.03; Fig. 4C–E). EFEMP1, SPON1, and CST3 remained significantly associated with the HHF endpoint after adjusting for both clinical risk factors and NT-proBNP.

Both trials featured repeat measurements with proteins assayed at baseline and a second timepoint (12 months for EXSCEL, 6 months for DECLARE). We evaluated proteins using the same procedure as with the baseline measurements (see Supplementary Fig. 19). In general, while the results were similar between time points, hazard ratios were systemically larger for the renal outcome in EXSCEL and the composite renal outcome in DECLARE (see Supplementary Fig. 20A and B). Additional proteins were significant for MACE, HHF, and the trial-specific respective renal outcome (e.g., TFF3) at the second timepoint. For the renal outcome in EXSCEL and the composite renal outcome in EXSCEL, this trend is partially ameliorated through the adjustment of baseline measurements (see Supplementary Fig. 20G and H). Overall, larger hazard ratios at the second time point could be indicative of cumulative exposure to a causal protein or reverse causality.

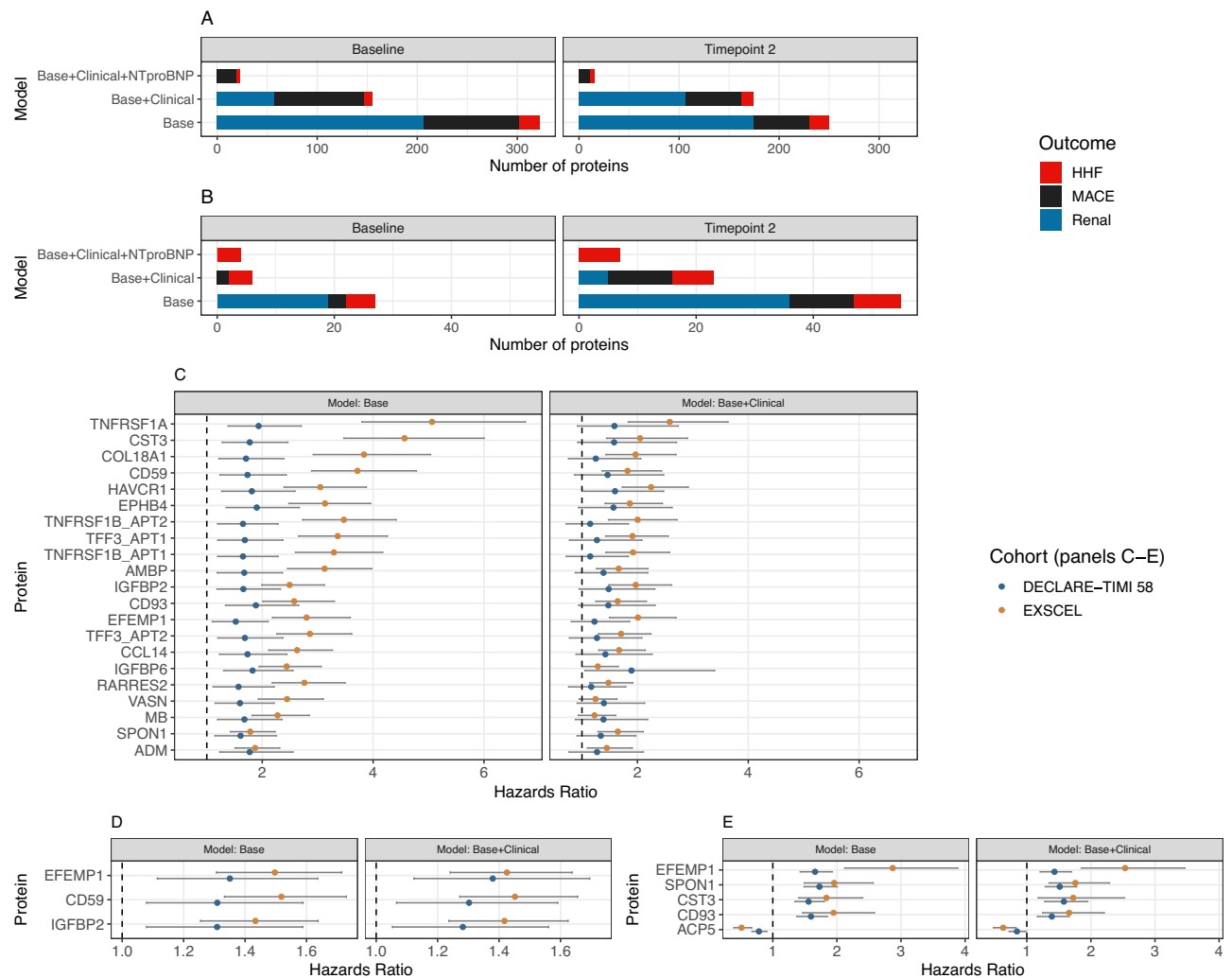

**Fig. 4 | Association of proteins with clinical trial outcomes and survival analyses. A** Summary of results in EXSCEL, with the *x*-axis corresponding to the three different models used (see Methods) and the *y*-axis corresponding to the number of proteins significant for each of the three outcomes (Bonferroni *p*-value < 0.05). **B** Summary of the replication results in DECLARE, with the *x*-axis corresponding to the three different models used and the *y*-axis corresponding to the number of proteins significant for each of the three outcomes (FDR *p*-value < 0.05). **C** Results from the survival analysis of the study-specific renal outcome in the placebo arms in EXSCEL and DECLARE. All displayed proteins replicated in DECLARE for the base model (with age, sex, age$^2$, age*sex, and genetic PCs 1–10 as covariates). **D** Results from the survival analysis of the major adverse cardiovascular event (MACE)

outcome in the placebo arms in EXSCEL and DECLARE. All displayed proteins replicated in DECLARE for the base model. **E** Results from the survival analysis of the hospitalisation for heart failure (HHF) outcome in the placebo arms in EXSCEL and DECLARE. For panels (**C–E**), the points represent the hazard ratios and error bars represent the 95% confidence interval obtained using the standard error for each hazard ratio. Note that all displayed proteins replicated in DECLARE for the base model. In panels (**C–E**), EXSCEL had a sample size of 1407 study participants from the placebo arm with available proteomics information, while DECLARE had a sample size of 497 study participants from the placebo arm with available proteomics information.

## PGS mediation in EXSCEL and DECLARE

Building on our endpoint analysis in the clinical trials, we sought to identify which protein biomarkers had evidence for causality. First, we evaluated whether any PGS were associated with clinical outcomes and, if so, whether they were associated with any circulating proteins in the trials (see Supplementary Data 16–18). We then performed mediation when both conditions were satisfied. In EXSCEL, we modelled $PGS_{CAD}$ as the exposure, $PGS_{CAD}$-associated proteins as the mediator, and MACE as the outcome. We found evidence that the $PGS_{CAD}$ mediates its effect through the circulating protein levels of C9, LBP, ITIH4, APOM, and HS6ST2 (Supplementary Fig. 21A and Supplementary Data 19). In DECLARE, the $PGS_{BMI}$ and $PGS_{CAD}$ were significantly associated with HHF and MACE, respectively. In the DECLARE mediation analysis, we did not identify any proteins that mediated the $PGS_{CAD}$'s effect. However, for the $PGS_{BMI}$ and HHF, we found evidence for 63 proteins mediating its effect (FDR-adjusted *p*-value < 0.05; see

Supplementary Fig. 21B and Supplementary Data 20). When including BMI in the models, 17 proteins still appeared to mediate the $PGS_{BMI}$ effect, though at a nominal significance level (*p* < 0.05; Supplementary Fig. 21C and Supplementary Data 20).

## Notable Pathways enriched across multiple PGS-protein sets

Next, to identify pathways underlying polygenic risk for cardiometabolic disorders, we performed pathway enrichment analysis of the PGS-associated proteins. The $PGS_{T2D\_gw}$ protein set was significantly enriched for 12 pathways after *p*-value adjustment (see **Methods**), of which 3 were also enriched in the $PGS_{CAD}$ protein set (Supplementary Fig. 22). Pathways that are significantly enriched in both the $PGS_{T2D\_gw}$ and $PGS_{CAD}$ protein sets may reflect shared causal mechanisms, such as those involving the complement system (Supplementary Figs. 23). Proteins in the complement and coagulation cascades pathway were identified as causal for cardiovascular disease in MR (C1R and C1S) and

mediation (18 complement-related proteins mediated the effect of PGS_{T2D_gw} and 4 complement-related proteins mediated the effect of the PGS_{CAD}). In the clinical trials, C9 mediated the effect of the PGS_{CAD} on the MACE outcome. In terms of biomarkers, C2 and CD59 from the complement cascade and PLAUR from the coagulation cascade were associated with the time to cardiovascular outcomes in EXSCEL and DECLARE (Supplementary Fig. 23B and D).

While the insulin-like growth factor binding proteins (IGFBPs) pathway was not enriched in the PGS_{CAD} or the PGS_{CKD} associated protein sets, it was enriched in the PGS_{T2D_gw}, PGS_{BMI}, and the PGS_{T2D_liver_lipid} protein association sets. In addition, its constituent proteins were identified in many of our analyses for both T2D and its comorbidities. IGFBP2 was among the most significant protein associations for the PGS_{T2D_gw}, while IGFBP4 and IGFBP6 were strongly associated with the PGS_{CKD} (Fig. 5A). While causal effects for IGFBPs on the tested diseases were not supported in our MR analysis, IGFBP2 and IGFBP6 were implicated with T2D and CKD, respectively, using mediation. In the clinical trials, many IGF-related proteins were significantly associated with both cardiovascular and renal outcomes (Fig. 5B and C).

## Discussion

In this study, we defined the intersection of polygenic risk for T2D, its cardiometabolic comorbidities, and the plasma proteome by

leveraging proteogenomic data across a population-scale biobank and two RCTs. As variants identified through T2D GWAS are often intergenic with small effect sizes, understanding their functional consequences can be challenging. By aggregating GWAS variants into PGS (including T2D, pathway-specific, and comorbidity PGS) and testing their association with large-scale proteomic data, our study provides an enhanced understanding of proteomic signature of T2D as well as causal proteins and pathways that may mediate T2D risk. Furthermore, our PGS-directed approach has the added benefit of evaluating more proteins for causality than analyses such as Mendelian randomisation (MR) that rely on individual variants (only 69% of the Olink-assayed proteins have sufficient pQTLs to perform MR). Our study has potential implications in highlighting biomarkers of T2D comorbidities, providing therapeutic targets for T2D, and uncovering the biological underpinning of cardiometabolic diseases.

Our study yielded insights into the proteomic consequences of polygenic risk for T2D and its comorbidities. The PGS_{T2D_gwas}, despite its derivation from the same summary statistics as the PGS_{T2D_gw}, was strongly associated with proteins identified by single locus methods, such as PAM, while the PGS_{T2D_gw}'s associations appeared to be more polygenic (Fig. 2). Given the causal links between adiposity and T2D, nearly all of the PGS_{T2D_gw} protein associations are at least partially mediated by BMI, though the contribution of BMI can range from relatively small (e.g., 7% of PGS_{T2D_gw}'s effect on MANSC4) to relatively

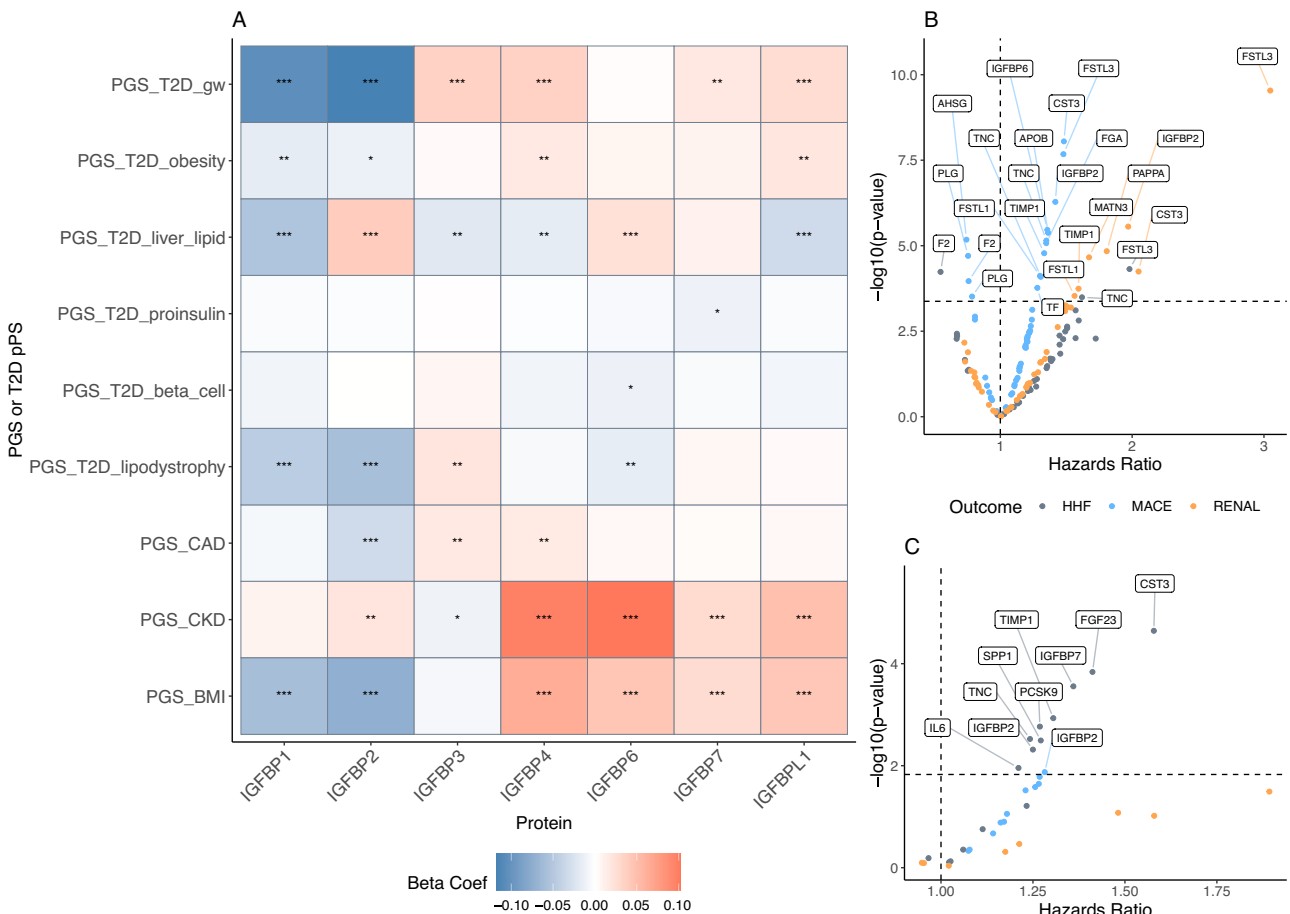

**Fig. 5 | IGF regulation by IGFBPs pathway. A** PGS associations with IGF binding proteins. A single asterisk (*) indicates the association was nominally significant, while two (**) indicates significance using FDR and three (***) indicates significance using a Bonferroni correction from linear regression of circulating protein levels in the UK Biobank. **B** Associations of proteins in this pathway with clinical trial outcomes in EXSCEL using Cox proportional hazards regression, adjusting for demographic covariates and clinical risk factors (see **Methods**). **C** Associations of

proteins in this pathway with clinical trial outcomes in DECLARE using Cox proportional hazards regression, adjusting for demographic covariates and clinical risk factors (see **Methods**). For panels **B** and **C**, the dashed line corresponds the p-value threshold where FDR < 5% (when applied to the proteins in this pathway). Note that panels (**B** and **C**) display unadjusted, two-sided p-values obtained from Cox proportional hazards regression.

large (e.g., 90% of the PGS$_{T2D\_gw}$'s effect on LEP). Notably, the partitioned T2D scores, apart from PGS$_{T2D\_obesity}$, were less influenced by adiposity (Supplementary Figs. 2–6). The partitioned T2D scores also seemed to capture unique aspects of T2D biology (189 proteins were associated with the partitioned scores but not the overall T2D score, i.e., PGS$_{T2D\_gw}$), thus, serving as proof of concept for the development of such scores. For example, the effect of the PGS$_{T2D\_beta\_cell}$ score on incident T2D was mediated by FAM3D, a causal relationship not detected by the overall PGS$_{T2D\_gw}$. FAM3D is thought to be involved in glucose regulation[29]. Further, we found the beta coefficients of PGS$_{T2D\_liver\_lipid}$ and the PGS$_{T2D\_gw}$ on the plasma proteome were frequently in the opposite direction (Fig. 2 and Supplementary Fig. 7). This is consistent with the known discordant effects of PGS$_{T2D\_liver\_lipid}$ and the PGS$_{T2D\_gw}$ on clinical biomarkers such as plasma triglycerides[13]. It seems plausible that the *GCKR* locus (index variant: rs1260236), captured by PGS$_{T2D\_liver\_lipid}$, drives a specific subtype of T2D with unique phenotypic consequences[13,15].

To identify PGS-protein associations that are attributable to forward causation, and discover potential therapeutic targets, we performed causal inference using Mendelian randomisation (MR) and mediation for the relevant clinical endpoints (i.e., T2D, BMI, and the T2D comorbidities CAD and CKD). For T2D, we found evidence using both MR and *cis* colocalization for 5 proteins, although two of them, ABO and APOE, likely harbour pleiotropic effects. PAM's involvement in the etiology of T2D has been described in previous studies[30]. Less is known about MANSC4 and NCR3LG1, but we note that MANSC4 has previously been identified via MR[31]. Among MR-significant proteins without colocalization evidence are NUCB2, ARG1, and LGALS4, all of which either play a role in glucose regulation[32], insulin resistance[33] or have been previously linked to diabetes[34,35]. For CAD, we identified 7 proteins with significant MR and *cis* colocalization evidence (see Fig. 3). Among these, LPA and PSCK9's role in the etiology of cardiovascular disease is well-established[36–38], while evidence in the literature for FURIN and FES continues to grow[39–41]. Notably, ERBB4, with significant MR estimates and *cis* colocalization for BMI (IVW: − 0.26, 95% CI: − 0.378 to − 0.138), has previously been linked to metabolic disorders and obesity[42–44]. With an alternative method, MR-Link-2, we identified additional proteins, such as FGF5 with CAD, and confirmed many of the associations we found with conventional MR. However, this approach also found more pleiotropic effects than our primary MR analysis (see Supplementary Data 9 and 10). As MR-Link-2 is designed to be pleiotropy-robust, we did not filter potentially pleiotropic instruments in advance as we did for the other MR analyses. In the case of some loci (such as LPA), the observed pleiotropic effect could be caused by statistically independent pQTLs in linkage disequilibrium[25].

Of the 56 proteins identified via MR, 40 were significantly associated with at least one PGS. For the three traits evaluated via mediation (CKD, CAD, T2D), 49% of MR-identified proteins also significantly mediated the effect of a PGS on incident disease. Proteins with causal evidence based on MR, promising as biomarkers for cardiovascular/renal comorbidities, or among the top 5 (by variance explained) of mediating proteins for each score are summarised in Supplementary Data 21, along with their druggability and gene expression information obtained from DrugnomAI[45,46].

We aimed to identify putative causal proteins and pathways that link T2D with its cardiovascular and renal comorbidities. To this end, we performed MR in patients with T2D as well as survival analysis and mediation analyses in EXSCEL and DECLARE-TIMI 58. With MR, we found that CCN4 (WISP1) and SELE conveyed risk for the development of cardiovascular outcomes while UMOD conveyed risk for renal comorbidities. UMOD's role in CKD has

previously been supported by MR[47], while CCN4 and SELE (E-selectin) are thought to be involved in T2D progression[48] and vascular inflammation[49], respectively. Several proteins implicated in our survival analysis, such as TFF3 and EFEMP1, could potentially explain mechanisms underlying the development of comorbidities[50,51]. We found 6 proteins that mediated the PGS$_{CAD}$'s effect on MACE in EXSCEL (C9, LBP, ITIH4, APOM, CES1, and HS6ST2), providing supporting evidence that they might serve as biomarkers for cardiovascular disease in patients with T2D[52–54].

Pathway enrichment of PGS-associated proteins can inform whether the proteins act collectively via certain biological pathways and can help guide therapeutic development (Supplementary Fig. 22). Pathways shared across multiple cardiometabolic PGS-protein sets could also reveal mechanisms relevant to the development of these comorbidities. Pathways involving the complement system, notably, were enriched in both the PGS$_{CAD}$ and PGS$_{T2D\_gw}$ protein sets. For example, C1R and C1S, which activate the C1 complex in the classical complement pathway[55], were significantly associated with the PGS$_{T2D\_gw}$ in addition to their MR and colocalization with CAD. Therapies targeting the complement system have been developed or proposed for a wide range of diseases, including inflammatory kidney disorders and cardiovascular disease[56,57]. The insulin-like growth factor binding proteins (IGFBPs) pathway was enriched in both the PGS$_{T2D\_gw}$ and PGS$_{BMI}$ protein sets. IGFBP2 was a biomarker for both cardiovascular and renal outcomes in EXSCEL and DECLARE-TIMI 58. Interestingly, lower levels of IGFBP2 are associated with incident T2D while higher levels are associated with incident CKD (see Fig. 5A), as has previously been observed[58,59]. While the exact physiological role of IGFBPs in the pathogenesis of cardiometabolic disease requires further investigation, we provided evidence of a shared role in both adiposity traits and cardiometabolic disease[60–62].

Our study has limitations. First, the trans-ancestry portability problem is commonly observed in biomedical research, including with PGS, though we sought to mitigate this by employing multi-ancestry PGS and trans-ancestry analyses. Second, since our study included UKB data, we selected scores that did not make use of the UKB (e.g., relatively older and smaller GWAS) to avoid overfitting, which may attenuate our theoretical statistical power. Third, despite excluding subjects with prevalent cardiometabolic diagnoses, the PGS-related analyses were not immune to reverse causality as several proteins strongly associated with the PGS$_{T2D\_gw}$ were implicated in our reverse MR analysis. It is also possible that we are capturing instances of feedback mechanisms. Fourth, defining phenotypes in the UKB via electronic health records could have an impact on analyses due to potential issues such as misclassification error[63]. Fifth, regression analyses including both T2D and BMI could exhibit collider bias as T2D and adiposity are causally linked, and as such, we do not consider any model that is significant after BMI adjustment when it is not significant without the BMI adjustment. Collider bias could also impact the analyses in the clinical trials as all participants were selected on a particular characteristic[64] (i.e., T2D with cardiovascular risk factors); however, adjusting for the risk factors as we have done could address this. Sixth, this study used data derived from circulating proteins, which may not necessarily represent the causative tissue for the diseases under examination. Seventh, while the cardiovascular outcomes in EXSCEL and DECLARE are aligned, the renal endpoints differ between the two trials (see **Methods**). This, in addition to differing eligibility criteria, could contribute to the effect size differences we observed in the survival analyses for the renal outcomes. Finally, the PGS, and by extension the mediation analyses using the PGS, could be impacted by horizontal pleiotropy as the PGS by design is not limited to variation reflecting a single exposure[9]. However, the partitioned scores help reduce this to a single mechanism. Our mediation sensitivity analyses

likely reflect this as many of our mediation models were not robust to potential mediator-mediator confounding (see **Methods**, Supplementary Data 12). The same limitation applies to MR analyses of complex traits and protein levels using *trans* pQTLs, though we have sought to minimise this confounding by testing for pleiotropy in our analytical framework.

Overall, we leveraged data from both a population-based setting and clinical trials to elucidate the proteomic signatures of polygenic risk for T2D and its comorbidities; provide causal evidence for the associated proteins; identify proteins and biological pathways that connect T2D with its comorbidities; and provide evidence for existing therapeutic and potentially new target opportunities. We also developed an interactive portal that allows users to interrogate and download the results of our analyses (https://public.cgr.astrazeneca.com/t2d-pgs/v1/).

## Methods

### Cohort description
The UK Biobank (UKB) is a deeply phenotyped population-based cohort comprised of approximately 500,000 subjects with array genotyping, exome/whole genome sequencing data and linkage to electronic health care record data with over 14 years of follow-up time[18]. The UK Biobank Pharma Proteomics Project (UKB-PPP) is a private-public partnership that assayed 2923 unique proteins (2922 after excluding one protein without sufficient measurements) in a subset of 54,306 UKB participants using the Olink Explore proteomics platform[16,17]. The UK Biobank operates under approval from the North West Multi-centre Research Ethics Committee (MREC). All UKB participants provided informed consent.

Exenatide Study of Cardiovascular Event Lowering (EXSCEL) examined the cardiovascular effects of once-weekly exenatide, a glucagon-like peptide-1 (GLP-1) agonist, in T2D patients with a median follow-up time of 3.2 years[19]. In a subset of trial participants ($N = 2823$), both genotyping and SomaScan proteomics data were generated (see Supplementary Data 1). All participants provided written informed consent, and the trial protocol was approved by ethics committees at each of the trial's participating sites. EXSCEL can be found on ClinicalTrials.gov (NCT01144338).

Dapagliflozin Effect on Cardiovascular Events–Thrombolysis in Myocardial Infarction 58 (DECLARE-TIMI 58) was a phase 3 RCT that examined the cardiovascular effect of dapagliflozin, an inhibitor of sodium-glucose co-transporter-2 (SGLT2), in patients with T2D with multiple risk factors for or established atherosclerotic cardiovascular disease with a median follow-up time of 4.2 years[20]. Similar to EXSCEL, for a subset of participants both genotyping and Olink proteomics data were available ($N = 915$). All participants provided written informed consent, and the trial protocol was approved by the institutional review board at each of the trial's participating sites. DECLARE-TIMI 58 can be found on ClinicalTrials.gov (NCT01730534).

See Extended Methods in **Supplementary Information** for a description of genotyping and proteomics quality control. Supplementary Fig. 1 shows the proteomic intersection between the three cohorts with proteomics data. We note that a previous study has found that SomaScan and Olink measurements to be moderately correlated[65].

### UK Biobank phenotype definitions
For the UKB, we used ICD10 codes and clinically meaningful "Union" phenotypes constructed by merging relevant ICD10 codes (release from Feb 2022) as detailed in Wang et al. [66]. For identifying prevalent cardiometabolic conditions that needed to be excluded from analyses, we used the following ICD10 codes: E10-E14 (any diabetes diagnosis), N18 (CKD), and I20-I25 (ischaemic heart disease, also known as CAD). Incident cases were defined using ICD10 codes (E11 for T2D, N18 for CKD, I25 for CAD) when the earliest date of diagnosis (determined by

fields 41270, 40001, 40002) occurred after baseline (when a sample was donated for proteomics). Note that relatively few cases were diagnosed within 30 days of sample collection (Supplementary Data 2).

### Randomised controlled trials (RCTs) phenotype definitions
We used three trial outcomes that were available in both trials, i.e., time to the composite cardiovascular outcome (major cardiovascular events or MACE, comprising of cardiovascular death, nonfatal myocardial infarction, or non-fatal ischaemic stroke), time to hospitalisation for heart failure (HHF), time to the renal outcome (for EXSCEL, two consecutive measurements of eGFR $< 30$ ml/min/1.73 m$^2$ and for DECLARE-TIMI 58, a composite renal outcome comprising of a sustained decrease of 40% or more in eGFR to $< 60$ ml/min/1.73 m$^2$, new end-stage renal disease, or death from renal causes). Note that the competing risk of death was addressed through censoring and the use of Cox regression[67]. DECLARE-TIMI 58 cardiovascular endpoints were adjudicated by independent adjudication committees. See Supplementary Data 3 for a description of phenotypes.

### Polygenic score estimation
To avoid overfitting when estimating the PGS in the UKB, we retrieved genome-wide association study (GWAS) summary statistics from external studies[43,68-73] that did not contain UKB participants (Supplementary Data 4), maximising both sample size and diversity. We trained genome-wide T2D, CAD, BMI, and CKD PGS using PRS-CS[74] (when only one GWAS per trait was available) or PRS-CSx[75] with these GWAS summary statistics. We ran PRS-CSx and PRS-CS with phi set to 'auto'. For the partitioned T2D polygenic scores (pPS), we obtained the variants and cluster weights generated by Udler et al. corresponding to five distinct genetic clusters, i.e., beta cell, lipodystrophy, liver lipids, obesity, and proinsulin[13]. To enable comparisons between genome-wide and GWAS-significant PGS, we also generated a set of GWAS-significant T2D summary statistics using the clump procedure implemented in PLINK v1.9 ($p$-value $< 5 \times 10^{-8}$, R$^2 < 0.1$, 250 kilobase window around each index variant). All PGS for UKB, UKB-PPP, EXSCEL, and DECLARE-TIMI 58 were estimated using post-QC imputed data and PLINK v2.00a4LM[76] (see Supplementary Data 5).

### PGS validation
To validate the PGS, we tested their associations with traits defined using ICD10 codes and/or biomarkers (Supplementary Data 2) in all unrelated UKB-PPP participants (resolved to the 2$^{nd}$ degree using KING 2.3.0[77]) and using R and adjusting for age, age$^2$, sex, age*sex, age$^2$*sex, UKB centre, array, and PCs 1-20. We also used KING 2.3.0, with the 1000 Genomes Project (1KGP), as a reference, to predict genetic ancestry. KING first performs principal component analysis on the 1KGP cohort, followed by projecting the UKB-PPP samples into the 1KGP PC space. Then, it uses a support vector machine and the 1KGP super-population labels to train a model and predict the ancestry of the target cohort. We tested the PGS for association with its target trait in the genetically predicted subsets (European, South Asian, East Asian, African, and Admixed American, all with a posterior probability $> 0.9$) and in the entire multi-ancestry cohort. For the CAD, BMI, and T2D, PRS-CSx inferred three sets of PGS (a "European" PGS, an "East Asian" PGS, and a meta-analysis of the two). In these cases, we retained the PGS with the strongest association with CAD, BMI, and T2D in the full cohort analysis.

### PGS and protein associations in UK Biobank
For testing PGS for association with protein expression levels, we modelled our analysis on the same internal replication structure as used in the UKB-PPP consortium pQTL GWAS. First, we restricted the analysis to unrelated participants (resolved to the 2$^{nd}$ degree) without a baseline diagnosis of diabetes or a relevant cardiometabolic condition (any Diabetes diagnosis, CAD, CKD) at data collection (UKB's baseline

timepoint; $N = 44,381$) to reduce confounding due to reverse causality as previously suggested by Ritchie et al. [9]. Then, we stratified the cohort into the consortium-identified discovery subset consisting of European-ancestry participants ($N = 29,496$) and the replication subset of remaining pan-ancestry participants ($N = 14,885$). We tested each PGS for association with protein expression levels in the discovery subset using linear regression in R, using the same covariates (age, age[2], sex, age*sex, age[2]*sex, batch, UKB centre, array, time to analysis, genetic PCs 1-20) as the UKB-PPP consortium pQTL GWAS[16]. Significant PGS-protein associations after a Bonferroni correction accounting for 10 scores and 2922 proteins ($p$-value $< 1.7 \times 10^{-6}$) were then moved forward to be tested in the replication subset. To account for the difference in sample sizes in the two subsets, we applied an FDR correction to our replication analysis separately to each PGS[9]. Associations were considered internally replicated with an FDR-adjusted $p$-value $< 0.05$.

To assess whether a PGS-protein association was driven by a single locus, we repeated the PGS-protein association analyses after adjusting for independent *cis* and *trans* pQTLs obtained from the UKB-PPP consortium's pQTL GWAS. Similarly, we repeated PGS-protein association analyses after adjusting for BMI, waist-hip ratio, and waist circumference to describe the influence of adiposity on the PGS associations. Finally, to explore the impact of genetic ancestry on the transferability of PGS-protein associations, we tested PGS-protein associations after stratifying the UKB-PPP cohort by the KING-predicted ancestry labels described above. We then compared and contrasted PGS-protein association patterns in the genetically predicted European ancestry subset with that of the other four predicted super-populations. As the UKB-PPP cohort is over 92% European ancestry, we used beta coefficients rather than $p$-values and compared the beta coefficients of the PGS on circulating protein levels using Pearson's r and the slope of the regression line fitted to the beta coefficients.

## Mediation of PGS-protein associations by adiposity

For the PGS$_{T2D\_gw}$-protein associations that were no longer significant, we performed mediation analysis to investigate if BMI mediated the effect of the PGS on the circulating protein levels. We used the 'mediate' function from mediation R package[78] (4.5.0) to perform mediation with the PGS as the exposure, BMI as the mediator, and the circulating protein levels as the outcome. We performed a sensitivity analysis using the 'medsens' function from the same R package.

## GWAS and pQTL summary statistics for Mendelian randomisation

We leveraged statistically independent pQTLs identified by the UKB-PPP consortium from the European ancestry 35K discovery subset to use as instruments for our two-sample Mendelian randomisation (MR) analysis. The UKB-PPP pQTL GWAS[16] used a minor allele count cut-off of 50, a minimum INFO score of 0.7, and a study-wide significant threshold $p$-value $< 3.4 \times 10^{-11}$. The pQTLs were defined as *cis* if they were within +/− 1MB of a gene's transcription start site. For this same set of pQTLs, we also extracted summary statistics using the pan-ancestry combined UKB-PPP cohort for our two-sample trans-ancestry MR analysis.

We then generated trait-level GWAS summary statistics using the UKB after excluding the UKB-PPP participants to avoid sample overlap. Individuals were retained for the GWAS if they passed an internal set of quality control criteria including sex concordance, heterozygosity, ploidy, were assigned a continental PEDDY-predicted[79] ancestry with a probability $\geq 0.90$, and were not included in the UKB-PPP study. For the generation of summary statistics for Mendelian randomisation, we used E11 for T2D, N18 for CKD, the union term of I20-I25 for CAD, and body mass index (BMI) measured at baseline. For traits in UKB patients

with T2D, we used the ICD10 codes in Supplementary Data 2, which contains a complete description of UKB phenotype definitions and cohort sample size (pre-QC). We utilised REGENIE for the GWAS, which employs a two-step approach as previously described[80]. For the first step, we used quality-controlled genotyped variants (MAF $> 1\%$, genotyping rate $> 99\%$, HWE $p$-value $> 10^{-15}$, $< 10\%$ missingness and LD pruning using 1000 variant windows, 100 sliding windows and $r^2 < 0.8$), while for the second step we used imputed variants with a MAC $> 50$ and an INFO score $> 0.7$. For traits assessed in the full cohort (T2D, BMI, CKD, CAD), we performed each GWAS within each of the predicted continental ancestries, adjusting for age, sex, and genetic PCs 1-10. For traits assessed in UKB participants with T2D, we performed each GWAS within the predicted European-ancestry subset, adjusting for age, sex, and genetic PCs 1-10.

## 2-sample Mendelian randomisation (MR)

We performed a two-sample Mendelian randomisation (MR) analysis to conduct a proteome-wide scan for proteins suspected to play a potential causal role in the development of T2D or T2D comorbidity. For our primary MR analysis, we utilised the European ancestry GWAS data. To identify weak instruments, we calculated the F-statistic[81] for each instrument (F-statistic = $\beta^2/Se^2$) and removed instruments with an F-statistic $< 10$. We also removed instruments that were pQTLs for 5 or more proteins to reduce pleiotropy. We performed MR on all proteins with 3 or more *cis* pQTLs using the MendelianRandomization[82] R package (version 0.7.0) and applying the simple and weighted median[83], IVW[84], and MR-Egger[85] methods. We then repeated the analysis using both *cis* and *trans* instruments. We addressed violations of the pleiotropy assumption by flagging results with an MR-Egger intercept $p$-value $< 0.05$[9]. We then took the median $p$-value across all MR methods to identify significant associations. For traits available in the entire UKB cohort, we reported both Bonferroni and FDR levels of significance, while for traits in patients with T2D (ICD10 code E11), we only applied an FDR correction due to the smaller sample sizes.

## Multi-variable Mendelian randomisation

As T2D and adiposity are closely intertwined, we performed multi-variable MR (MVMR) using the MVMR R package[26], BMI GWAS data from the FinnGen cohort (r11)[86] and pQTL data from the UKB-PPP discovery cohort (European ancestry) as exposures, and T2D GWAS data from the European-ancestry subset of the UKB (excluding UKB-PPP participants) as the outcome. The FinnGen study is a large-scale genomics initiative that has analyzed over 500,000 Finnish biobank samples and correlated genetic variation with health data to understand disease mechanisms and predispositions. The project is a collaboration between research organisations and biobanks within Finland and international industry partners. MVMR reports both instrument strength and pleiotropy estimations, and we retained analyses with the conditional F-statistic $> 10$ for both exposures and no evidence of pleiotropy (Q-statistic $p$-value $> 0.05$). Like our primary MR analysis, we repeated MVMR using *cis* only and *cis* plus *trans* instruments. We then applied an FDR correction on the results.

## Mendelian randomisation using cis molecular data and MR-Link-2

We also performed *cis* MR analysis using MR-Link-2[25], a methodology developed explicitly for MR using GWAS information obtained *cis* molecular data. We again use the pQTL GWAS data from the UKB-PPP as the exposure and the UKB GWAS data as the outcome, though in this case taking the entire *cis* region as input. MR-Link-2 requires genotype data for LD, and for this, we used the UKB-PPP participants from the GWAS data (the European ancestry discovery cohort).

**Mendelian randomisation sensitivity analyses including statistical colocalization**

For sensitivity analyses, we repeated MR analyses using pan-ancestry summary statistics from the full UKB-PPP cohort and the meta-analysed trans-ancestry GWAS MR analyses (via a fixed-effected meta-analysis). We also conducted a colocalization analysis for any variant with GWAS $p$-value $< 1 \times 10^{-6}$ for both the trait and the protein using the coloc package[87] (version 5.1.0.1) in R 4.2.2 on a +/− 250 kilobase window centred on the variant. We used the summary statistics-based method (coloc.abf) with the default priors (p1: $10^{-4}$, p2: $10^{-4}$, p12: $10^{-5}$) and considered the GWAS variant and the pQTL to have strong colocalization evidence following recommended criteria (PP.H3 + PP.H4 > 0.99 and PP.H4/PP.H3 > 5)[9,87,88]. For loci located in the *cis* region of an Olink-assayed protein, we also used the coloc.susie[89] method with LD matrices computed with the imputed UKB-PPP data and PLINK v1.90b6.18[76,90].

**Mediation analyses of PGS, circulating proteins, and incident disease in UKB**

In this framework, we set the PGS as the exposure, the protein as the mediator, and incident disease as the outcome. We utilised incident cases (prevalent cases were excluded), the same set of covariates as the UKB PGS-protein association analysis (see above), and the Medflex R package[9,91] (version 0.6-10) to perform mediation analysis with natural effects models. We considered a mediation model to be significant if the mediation (indirect) $p$-value and the total $p$-value were both significant after a Bonferroni correction ($p$-value $< 1.4 \times 10^{-6}$); the direct effect $p$-value was allowed to be not significant as it is possible that the PGS's effect is primarily mediated through the tested protein. As a sensitivity analysis, we performed mediation for both all participants and only European ancestry; we filtered out proteins that were not significant in both scenarios.

To evaluate robustness to potential mediator-mediator confounding, we employed the mediation R package (4.5.0)[78]. First, we again performed mediation with the PGS as exposure, the protein as a mediator, and the incident disease as the outcome using the 'mediate' function. Then, we performed the sensitivity analysis using the 'medsens' function, which performs a simulation-based sensitivity analysis. As the PGS is fixed at birth, it is assumed that there will not be any PGS-mediator confounding.

**Reverse Mendelian randomisation**

We obtained MR instruments for reverse MR from the European-ancestry T2D, CKD, BMI, and CAD GWAS data that we generated for our forward MR analysis, plus an additional model for CAD using only the ICD10 code I25. We performed LD pruning using PLINK v1.90b6.18's clumping procedure with a 500 KB window size, a minimum $p$-value threshold of $5 \times 10^{-8}$, and R² < 0.001 using UKB-PPP as the LD reference. We matched each instrument with variants from the pQTL GWAS data. We then performed reverse MR using the MendelianRandomization R package in the same manner as described above.

**Association and mediation of proteins with trial outcomes in EXSCEL and DECLARE-TIMI 58**

We tested proteins for their association with the time to EXSCEL outcomes with the survival package in R 3.6.1 (https://github.com/therneau/survival) and replicated significant associations in DECLARE-TIMI 58. In both cases, we restricted analyses to the placebo arm. We used Cox proportional hazards regression and adjusted for age, sex, age*sex, and genetic PCs 1-10. For both EXSCEL and DECLARE-TIMI 58, two time points were available for proteomics: baseline plus 12 months for EXSCEL, and baseline plus 6 months for DECLARE. In both studies, we performed the proportional hazards regression analysis three times, using the baseline measurements, the repeat measurement, and the repeat measurement with baseline included as an additional covariate, respectively, as exposures. For EXSCEL, each SomaLogic aptamer was tested separately and reported. In the case of statistically significant associations in the clinical trial time-to-event analyses, (FDR $p$-value < 0.05 as well as a more stringent Bonferroni-corrected $p$-value in EXSCEL), we assessed the proteins using more comprehensive models that included clinical risk factors to evaluate their suitability as a biomarker (see "Description of risk factors" section below). For all models, we tested for the proportional hazards assumption using the cox.zph() function and flagged any model with a global $p$-value < 0.05 as violating model assumptions.

**Description of risk factors included in clinical trial time-to-event analyses**

In the case of statistically significant associations in the clinical trial time to event analyses, we assessed the proteins using more comprehensive models to evaluate their suitability as a biomarker. These models included both the above covariates along with additional risk factors specific to the outcome of interest. For heart failure outcomes, we also adjusted for coronary artery disease, atrial fibrillation, baseline eGFR, baseline UACR (diagnosis of albuminuria for EXSCEL), and prior heart failure[92]. For MACE, we also adjusted for hypertension, hypercholesterolaemia, BMI at baseline, smoking (current, or smoking cessation ≤3 months), and atherosclerotic disease (prior myocardial infarction, percutaneous coronary intervention/coronary artery bypass grafting, cerebrovascular accident/transient ischaemic attack, or peripheral arterial disease)[93]. Note, for DECLARE, the prior cardiovascular disease and multiple risk factors variables were used as they captured the MACE risk factors. For renal outcomes, we adjusted for atherosclerotic cardiovascular disease at baseline, heart failure at baseline, systolic blood pressure at baseline, T2D duration, HbA1c at baseline, eGFR at baseline, urine albumin-to-creatinine ratio (ACR) at baseline, and haemoglobin at baseline[94]. Note, for EXSCEL, certain analytes were only assessed intermittently, resulting in sample size loss when adjusting for these covariates. Consequently, we replaced ACR with a baseline diagnosis of albuminuria and performed a sensitivity analysis with and without haemoglobin. Despite the substantial difference in sample sizes, the hazard ratios were highly correlated ($r = 0.91$, $p$-value $< 2.2 \times 10^{-16}$), thus we omitted the haemoglobin adjustment for EXSCEL. Finally, for HHF and MACE, we included a third model that also adjusted for NT-proBNP to identify proteins that were independent of this biomarker as NT-proBNP was profiled in both EXSCEL and DECLARE.

**Mediation analysis in EXSCEL and DECLARE**

In the scenarios where a PGS is associated with both a clinical trial outcome and a protein (see **Extended Methods**), and the protein is in turn also associated with the same outcome, we performed mediation analysis using the Medflex R package (version 0.6-10) in the same manner as above, with the indirect $p$-value adjusted using the FDR approach. To maximise sample size for mediation, we made use of the full proteomics cohort and included the treatment arm as a covariate in addition to age, sex, age*sex, and genetic PCs 1-10.

**Pathway enrichment**

For all sets of proteins identified by the PGS-protein analyses, we used the gProfiler[95] tool (https://biit.cs.ut.ee/gprofiler/gost) to test if these sets of encoded genes were enriched for KEGG, WikiPathways, or REACTOME pathways. Note that we restricted the statistical domain to the genes whose protein products are captured by the Olink panels.

**Reporting summary**

Further information on research design is available in the Nature Portfolio Reporting Summary linked to this article.

## Data availability

The data generated by this study (filtered for a nominal *p*-value of < 0.05) are available in the supplementary materials, many of which are also available via the web portal. Source data for the display figures has been provided. The UK Biobank has assigned the proteomics dataset to Category 1839 and 'Field 30900', details of which can be found here: https://biobank.ndph.ox.ac.uk/showcase/label.cgi?id=1839. Requests to access UK Biobank data can be made here: https://www.ukbiobank.ac.uk/enable-your-research/apply-for-access. Clinical trial data can be accessed following AstraZeneca's data-sharing policies: https://www.astrazenecaclinicaltrials.com/our-transparency-commitments/. The PGS used in the study are available in the PGS Catalogue (https://www.pgscatalog.org/) under Publication ID PGP000701 and score IDs PGS005110-PGS005119. Source data are provided in this paper.

## Code availability

This study was carried out using only publicly available software. Analysis scripts are available at https://github.com/astrazeneca-cgr-publications/plasma-proteomic-markers-prs-t2d-scripts.

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

## Acknowledgements

We thank the participants and investigators of the UK Biobank study who made this work possible (Resource Application Numbers 26041 and 65851). We are grateful to the research and development leadership teams at the 13 participating UKB-PPP member companies (Alnylam Pharmaceuticals, Amgen, AstraZeneca, Biogen, Bristol-Myers Squibb, Calico, Genentech, Glaxo Smith Klein, Janssen Pharmaceuticals, Novo Nordisk, Pfizer, Regeneron, and Takeda) for funding the study. We also want to acknowledge the participants and investigators of DECLARE-58 TIMI, EXSCEL, and the FinnGen studies. M.I. and S.C.R. were supported by core funding from the British Heart Foundation (RG/18/13/33946) NIHR Cambridge Biomedical Research Centre (BRC-1215-20014; NIHR203312) [*]. M.I. was also supported by the Cambridge BHF Centre of Research Excellence (RE/18/1/34212), BHF Chair Award (CH/12/2/29428) and by Health Data Research UK (Molecules to Health Records programme), which is funded by the Medical Research Council (UKRI), the National Institute for Health Research, the British Heart Foundation, Cancer Research UK, the Economic and Social Research Council (UKRI), the Engineering and Physical Sciences Research Council (UKRI), Health and Care Research Wales, Chief Scientist Office of the Scottish Government Health and Social Care Directorates, and Health and Social Care Research and Development Division (Public Health Agency, Northern Ireland. *The views expressed are those of the authors and not necessarily those of the NIHR or the Department of Health and Social Care.

## Author contributions

D.P.L., A.N., D.S.P., S.C.R. and M.I. conceptualised this study. D.P.L. performed analyses and drafted the main text and supplementary materials. A.N., D.S.P., and M.I. supervised the study. D.M. and D.V. developed the web portal. M.G. and X.J. assisted with data curation. B.B.S., H.R., C.D.W. and S.P. contributed to the development of the UKB-PPP resource. R.J.M. and R.R.H. led the EXSCEL clinical trial. F.A.M., S.D.W. and M.S.S. led the DECLARE-TIMI 58 clinical trial (including several post-hoc analyses) and/or genotyping and proteomics data collection. M.S.U. developed the partitioned T2D polygenic scores. I.A.G.-N. and J.O. provided insight into the DECLARE-TIMI 58 trial and T2D biology. All authors read, commented on, and agreed upon the submitted manuscript.

## Competing interests

D.P.L., M.G., D.M., D.V., X.J., I.A.G., S.P., J.O., A.N. and D.S.P. are employees of AstraZeneca and may hold AstraZeneca stock options. B.B.S. and H.R. are employees of Biogen and may hold stock options. C.D.W. is an employee of Janssen Pharmaceuticals, a Johnson & Johnson company, and may hold stock options. R.R.H. reports personal fees from Anji Pharmaceuticals, AstraZeneca and Novartis. R.J.M. received research support and honoraria from Abbott, American Regent, Amgen, AstraZeneca, Bayer, Boehringer Ingelheim, Boston Scientific, Cytokinetics, Fast BioMedical, Gilead, Innolife, Eli Lilly, Medtronic, Medable, Merck, Novartis, Novo Nordisk, Pfizer, Pharmacosmos, Relypsa, Respicardia, Roche, Rocket Pharmaceuticals, Sanofi, Verily, Vifor, Windtree Therapeutics, and Zoll. M.I. is a trustee of the Public Health Genomics (PHG) Foundation, a member of the Scientific Advisory Board of Open Targets and has research collaborations with Nightingale Health and Pfizer which are unrelated to this study. F.A.M. received consulting fees from Janssen. S.D.W. received grants from Amgen, AstraZeneca, Daiichi Sankyo, Eisai, Janssen, Merck, and Pfizer, and consulting fees from AstraZeneca, Boston Clinical Research Institute, Icon Clinical, and Novo Nordisk. M.S.S. received research grant support through Brigham and Women's Hospital from Abbott, Amgen, Anthos Therapeutics, AstraZeneca, Boehringer Ingelheim, Daiichi-Sankyo, Eisai, Intarcia, Ionis, Merck, Novartis, and Pfizer, and consulting for Althera, Amgen, Anthos Therapeutics, AstraZeneca, Beren Therapeutics, Boehringer Ingelheim, Bristol-Myers Squibb, Dr. Reddy's Laboratories, Fibrogen, Intarcia, Merck, Moderna, Novo Nordisk, Precision BioSciences, and Silence Therapeutics. The remaining authors declare no competing interests.

## Additional information

[1]Centre for Genomics Research, Discovery Sciences, BioPharmaceuticals R&D, AstraZeneca, Cambridge, UK. [2]British Heart Foundation Cardiovascular Epidemiology Unit, Department of Public Health and Primary Care, University of Cambridge, Cambridge, UK. [3]Victor Phillip Dahdaleh Heart and Lung Research Institute, University of Cambridge, Cambridge, UK. [4]British Heart Foundation Centre of Research Excellence, University of Cambridge, Cambridge, UK. [5]Health Data Research UK Cambridge, Wellcome Genome Campus and University of Cambridge, Cambridge, UK. [6]Cambridge Baker Systems Genomics Initiative, Department of Public Health and Primary Care, University of Cambridge, Cambridge, UK. [7]Translational Sciences, Biogen Inc., Cambridge, MA, USA. [8]Data Science, Janssen Pharmaceuticals, Princeton, NJ, USA. [9]Diabetes Trials Unit, Radcliffe Department of Medicine, University of Oxford, Oxford, UK. [10]Division of Cardiology, Duke University School of Medicine, Durham, NC, USA. [11]Thrombolysis in Myocardial Infarction (TIMI) Study Group, Division of Cardiovascular Medicine, Brigham and Women's Hospital and Harvard Medical School, Boston, MA, USA. [12]Section of Cardiovascular Medicine, Department of Internal Medicine, Yale School of Medicine, New Haven, CT, USA. [13]VA Connecticut Healthcare System, West Haven, CT, USA. [14]Diabetes Unit, Massachusetts General Hospital, Boston, Massachusetts, USA. [15]Center for Genomic Medicine, Massachusetts General Hospital, Boston, Massachusetts, USA. [16]Broad Institute of MIT and Harvard, Cambridge, Massachusetts, USA. [17]Department of Medicine, Harvard Medical School, Boston, Massachusetts, USA. [18]Late-Stage Development, Cardiovascular, Renal and Metabolism, BioPharmaceuticals R&D, AstraZeneca, Gothenburg, Sweden. [19]Precision Medicine and Biosamples, Oncology R&D, AstraZeneca, Cambridge, UK. [20]Cambridge Baker Systems Genomics Initiative, Baker Heart and Diabetes Institute, Melbourne, Victoria, Australia. [21]These authors jointly supervised this work: Abhishek Nag, Dirk S. Paul, Michael Inouye. ✉e-mail: douglas.loesch@astrazeneca.com; dirk.paul@astrazeneca.com

