## [Transparent Peer Review file · Nature Communications]

Identification of plasma proteomic markers underlying polygenic risk of type 2 diabetes and related comorbidities

Corresponding Author: Dr Douglas Loesch

Version 0:

Reviewer comments:

Reviewer #1

(Remarks to the Author)

The study by Douglas P. Loesch et al. provides valuable insights into the complex pathophysiology of type 2 diabetes (T2D) and its associated comorbidities using a powerful proteogenomic approach. The authors leveraged the UK Biobank, a large population-based cohort, and two randomized controlled trials (EXSCEL and DECLARE-TIMI 58) to identify circulating protein biomarkers associated with polygenic risk scores (PGS) for T2D and related cardiometabolic conditions. The study's strengths include the large sample size, the use of multi-ancestry participants, and the validation of findings in independent clinical trial cohorts.

The authors identified a large number of proteins (839 for T2D PGS and 1,005 for cardiometabolic PGS) associated with genetic risk for T2D and its comorbidities. Some of these proteins, such as TFF3, EFEMP1, and MMP12, were further associated with renal and cardiovascular outcomes in the clinical trials, suggesting their potential as prognostic biomarkers. The study revealed shared pathways underlying T2D and its comorbidities, including the complement cascade, cholesterol metabolism, and IGF signaling.

Major comments:

The authors tackle an important topic and make use of valuable resources, especially those from EXSCEL and DECLARE-TIMI 58 trials, which are not publicly available. I appreciate the extensive analyses the authors have performed and their effort to make the data accessible through their web portal. To further enhance the clarity, coherence, and impact of the work, I would like to offer the following suggestions.

1. Clarifying the main purpose and key findings: I suggest the authors refine the narrative, text, and figures to more clearly convey the study's main objectives and the most important results. This will help readers better appreciate the significance of the work.
2. Clarifying the connection between UKB and clinical trial analyses: To improve the clarity of the link between the analyses in UKB and the two clinical trials, I would recommend the authors elaborate on the logic, e.g., "We conducted extensive polygenic score (PGS), Mendelian randomization (MR), and mediation analyses in UKB to identify highly probable potential biomarkers. We then tested the associations between these biomarkers and cardiovascular events in the two clinical trials, replicating the N[number] protein-trait associations out of M[number] (FDR < XXX)."
3. Aligning the outcome traits between the analyses: Considering that EXSCEL and DECLARE-TIMI 58 were primarily focused on cardiovascular outcomes in individuals with T2D, I would suggest the authors focus on cardiovascular events as the outcomes in the analyses in UKB rather than T2D itself or BMI. In addition, blood or plasma may not be the primary causal tissue for T2D and BMI.
4. Highlighting important findings and their logic: Figures 3 and 4, which show the consistent association of IGFBPs with renal outcomes and major adverse cardiovascular events (MACE), appear to be crucial findings. I suggest the authors more clearly present the logic with which they narrowed down to this pathway through extensive analyses in UKB and clinical trials. If some analyses are not relevant to the main findings, they could be less emphasized or even omitted from the main texts.
5. Streamlining supplementary materials: Given the extensive number (more than 60) of supplementary figures and tables, I

suggest the authors select the important ones (less than 30 in total, for example) and simplify the overall logic of the study. Providing a table or a diagram summarizing which protein-trait relations are supported by each analysis could be helpful.

6. Improving transparency: I recommend clearly stating the thresholds of significance of each analysis in the main text.

Minor comments:

- Line 45: "liver lipid" is duplicated.
- Line 54: "Some PGS-associated proteins ...". I suggest specifying the number of PGS-associated proteins that were associated with renal and cardiovascular trial outcomes.
- Line 359: To maintain consistency in target phenotypes and improve the rationale of the paper, I recommend considering the removal of diabetic retinopathy and hypertension from the causal inference using MR.
- Lines 368-369: I suggest clarifying whether the analysis using all diabetes-related ICD-10 codes (E10-E14) provided any additional findings compared to the analysis using only T2D (E11). If not, please consider removing the E10-E14 analysis, as type 1 diabetes (E10) and T2D (E11) are different diseases.
- Figure 4 B and D: The connection between plots and the protein names are vague. Please consider drawing lines between plots and protein names.

(Remarks on code availability)

Reviewer #2

(Remarks to the Author)

This study investigated polygenic scores (PGS) for type 2 diabetes (T2D), for five clusters of T2D loci indicating genetic heterogeneity in T2D, and for selected T2D comorbidities – i.e., coronary artery disease (CAD), chronic kidney disease (CKD), body mass index (BMI), and non-alcoholic fatty liver disease (NAFLD) – for associations with close to 3,000 proteins in over 40,000 men and women in UK Biobank. The authors then tested the PGS-associated proteins for associations with incident cardiometabolic complications – i.e., insulin initiation, major cardiovascular events, hospitalization for heart failure, and renal complications – in two cardiovascular outcome trials among patients with T2D in a prospective analysis of data from two randomized controlled trials.

I think this study addressed a very relevant and important research question using an elaborate study design and applied solid statistical approaches. The work has the potential to contribute substantially to the field and can also inspire related field. The manuscript is largely well written; however, it was sometimes difficult to follow. Due to the complex design, it was not always easy to follow which exposure was linked to which outcome. I have several suggestions that may help to improve the accessibility of the manuscript. I also have several comments regarding the methods, which may need clarification.

- 1) Abstract: It would be helpful to list the T2D associated comorbidities that were studied in the abstract. Indicate in the abstract whether the proteins associated with T2D PGS were significant after multiple testing correction.
- 2) First paragraph of the introduction: Can the author give one example of how genome-wide association studies (GWAS) have identified genetic associations that have advanced our understanding of pathophysiological pathways underlying T2D? In the same sentence, can authors elaborate very briefly why connecting individual T2D risk variants to specific pathways remains challenging and how their study may address this challenge.
- 3) Of the overlapping proteins measured across the two platforms (Olink and SomaScan), can the authors briefly elaborate on the comparability of measurements after normalization (e.g., in terms of abundance levels).
- 4) Please clarify in the last paragraph of the UK Biobank phenotype definitions section whether the generation of summary statistics for the Mendelian randomization analysis are already published GWAS results, or it was performed by the authors. If they are published, more information from where they were extracted is needed.
- 5) Provide abbreviations when first mentioned in manuscript (e.g. hospitalization for heart failure); also provide abbreviations in footnotes of all tables and figures.
- 6) It would be appreciated to present the correlations between the different PRSs on T2D, CAD, BMI, CKD, and the partitioned T2D PS. Also, between the genome-wide and GWAS-significant models.
- 7) I was wondering what the rationale was of choosing PRS-CS/PRS-CSx to derive the PRSs. Did the authors compare the performance with other software such as LDpred2-auto which showed to overcome PRS-CS?
- 8) Where potential non-linear associations between PGS and proteins investigated – please clarify and if not, why not. Please clarify whether phenotypical or PGS BMI was used for adjustment in PGS-protein associations.
- 9) I found it a bit confusing how you predicted the ancestries groups (end of PGS and protein associations in UK Biobank section). I suggest giving more detail on it. Was it based on the genetic principal components?
- 10) It is confusing what the authors mean by this sentence: "We then compared the effect sizes (beta coefficients) of the PGS

on circulating protein levels between ancestry groups using Pearson's r and the slope of the regression line fitted to the beta coefficients." A clarification would be informative for the reader. Same is applicable for the section where these results are presented (page 15). Also, what was the proportion of different ancestries in the population?

11) In describing the mediation analysis, please clearly describe the exposure, the mediator, and the outcome. Also, at which level the exposure and mediator were modelled. Were proteins modelled in separate models or mutually adjusted for? How were mediator-mediator confounding and exposure-mediator confounding treated? A DAG would be helpful.

12) In the 2-sample Mendelian randomization (MR) section further details are needed. It is not clear from where the data was extracted nor how many participants included the GWAS used in the analysis. How many genetic variants were included to build the genetic instruments? Was any filter applied to strengthen the genetic instrument (e.g., LD, R^2 ...)? Also, are both samples (exposure and outcome) independent and from same population ancestries (these should be clarified to have insights of potential violation of the second MR assumption). I strongly suggest following the MR-STROBE available here: <https://www.strobe-mr.org/>. Additionally, please add the references for the several MR methods used.

13) Cox PH regression: Please indicate the time scale of analysis; list all covariates either in the main text or if word count is an issue, in the footnotes of relevant tables and figures; where PH assumptions tested and satisfied? I think interactions between trial arm and proteins should be accounted for.

14) Briefly describe the different study populations.

15) Page 13, first two sentences of results can rather be moved to into introduction or discussion; an understanding of the study population would be more informative.

16) When reporting results, clarify whether reported associations were observed after multiple testing correction.

17) I find it confusing to say that protein effect sizes were correlated – effect size is not a unit of a variable. I guess that beta coefficients were extracted and correlated? Please clarify what was done, which correlation method was used, and modify throughout the manuscript.

18) Please add the estimate and confidence interval in the results for ERBB4 in page 16.

19) First paragraph of discussion could rather go to the second paragraph of discussion with relevant references.

20) It is not always clear to the reader when authors talk about mediation results in terms of exposure, mediator, and outcome. E.g., second paragraph of the discussion, FAM3D mediated which relationship?

21) First sentence, fourth paragraph: it is unclear what is meant with "inverse relationship between T2D and CKD risk" – were associations between T2D and CKD risk investigated?

22) Figure 3, Since HR at months 12 are almost consistently larger, can this indicate reverse causation? Did the authors consider cumulative exposure to proteins, e.g., by adjusting for baseline. What would complete concordance tell us?

23) Figure 4, Kaplan-Meier is little informative - one cannot distinguish the two groups.

24) Use person first language throughout the manuscript including supplementary material: e.g., patients with T2D or adults with T2D instead of "diabetics" or "T2D patients".

(Remarks on code availability)

Reviewer #3

(Remarks to the Author)

The paper by Loesch DP E et al examined the causal associations of circulating proteins with polygenic scores for T2D and its main cardiometabolic complications, utilising newly emerging Olink proteomics data in the UK Biobank, with further follow-up replications and explorations of findings in two randomised trials. T2D is a common but highly heterogeneous conditions. Although a large number of genetic variants have been identified, the biological mechanisms underlying T2D genetic risks and the molecular characteristics of different genetic subtypes of T2D defined by polygenic scores (e.g. beta cell, lipodystrophy, obesity, liver lipid, and proinsulin) remain poorly understood. The present study represented the largest investigation to date on plasma protein markers underlying genetic risk of T2D and the related complications, using a range of analytic methods (e.g. MR, mediation, pathway enrichment analyses). The study designs are appropriate, analytic methods and procedures robust, and study limitations well recognised. Importantly, the study has generated many informative and novel findings that have the potential to improve understanding of T2D aetiology and inform risk prediction, patient stratification and development of new treatments. However, there are several general and specific issues related to data analyses, presentation and interpretation, which could be further clarified and improved.

1. In various analyses, BMI was adjusted. While appropriate, it would be helpful to also consider adjusting for other

measures of adiposity, particularly WC and WHR. As measures of central obesity, WC and WHR have been shown in both conventional and genetic analyses to be more strongly associated with risk of T2D compared with BMI. Moreover, there is emerging evidence from recent proteomics-related research that the disease-causing pathways associated with central obesity differ somewhat from those associated with BMI. Hence, it would be helpful to look into this separately and compared the findings with those related to BMI.

2. Examine and clarify the effects of potential collider/selection bias in the following scenarios:

- a. Adjusting for BMI when assessing the association of PGS with proteins, in which BMI can be considered as a mediator between PGS and proteins, leading to potential collider bias.
- b. Applying BMI-adjusted GWAS for the MR analysis would be subject to collider bias. MVMR would be an alternative method, although instrument used may be weaker (the conditional F statistics could be checked beforehand).
- c. When assessing the association of PGS with proteins in the two RCT patient populations (EXSCEL and DECLARE-TIMI 58), T2D is likely a mediator between PGS associated proteins and clinical trait outcomes, although the 3 models might address the confounders.

It would be helpful to undertake additional analyses and to discuss their likely impacts and acknowledge these as potential study limitations.

3. It is not very clear as to why the median of all the MR estimates (IVW, weighted IVW, median, and Egger methods, as well as the p-value and 95% CI) was used to interpret the MR results. Each method has its assumptions for specific purpose. Simplifying all the estimates into one median estimate will likely lose information. For example, an "outlier" MR estimate may provide specific insights in some cases. Besides, the median estimate would be driven by how many MR methods that have been applied. It would be helpful to provide further details in interpreting the MR findings based on ensemble MR estimate.

4. Likewise, it is not clear as to why the MR analyses were confined only to those proteins with ≥ 3 cis-/trans-pQTLs. Although trans-pQTLs need to be handled with caution, there is currently no general consensus as to how the cis- and trans-pQTLs should be incorporated into a single MR estimate. Besides, some novel methods have recently been proposed (e.g. MR-Link-2, MR-BMA), which may provide a useful alternative to ensemble MR approach, which could be over-simplified. Moreover, to help establish causality reliably, it may be more appropriate to use the ensemble MR estimate only for screening purpose. But still, it is necessary to triangulate all the MR (including all pleiotropy robust estimates, cis-pQTLs-based MR estimates, and cis- and trans-pQTLs-based MR estimates) and the colocalization evidence on the candidate proteins obtained from the screening step.

5. While appropriate and informative, the reverse MR seemed to have involved all rather than only PGS-associated proteins. As such the study findings may be subject to reverse causality, for the proteins caused by specific diseases should not be considered as the outcomes in this particular setting.

6. Given the range of analyses undertaken and study findings generated, it would be helpful to provide a summary table for all or selected top causal proteins identified, showing all the key findings, including tissue expression, secreted or not, evidence of drug development, overlap across different disease outcomes, etc. This should help readers to fully understand and appreciate the breadth and depth of the findings and opportunities for further translational research.

7. While all the key elements are properly described and discussed, the writing could be further restructured and improved. For example, in the Results section, there are many undue repetitive justifications for specific analyses and use of certain datasets (e.g. lines 3-6 in Page 13; lines 15-18 in Page 17; lines 14-18 in Page 19), which are properly described in the Methods section. Similarly, there are many undue discussion of the study findings in the Results section (e.g. last three lines in Page 13), which makes the Discussion section rather weak and lightweight. This is particularly the case for the Results section related to two randomised trials, which could be shorten significantly without losing any key messages. In discussion section, it would be helpful to highlight briefly the study strengths in the context of existing knowledge and research methods, prior to discussion of its limitations.

(Remarks on code availability)

Reviewer #4

(Remarks to the Author)

(Remarks on code availability)

Reviewer #5

(Remarks to the Author)

(Remarks on code availability)

NA

Version 1:

Reviewer comments:

Reviewer #1

(Remarks to the Author)

I thank the authors for addressing comments from me and other referees. The manuscript has been significantly improved since the previous submission. To help further refine the work, I've outlined some suggestions below:

1. Abstract: The abstract could be further refined to better reflect the study's key findings and methodologies. The introductory sentences (lines 38-43) may be a bit wordy and can be condensed. Please consider describing the downstream analyses such as causal inference, pathway enrichment, and Cox regression more in detail.

Minor comments:

2. Figure 2C: please consider changing "T2D_PGS" to "PGS_T2D_gw" for consistency and clarity.

3. Figure 4: The current title "Survival analysis in the clinical trials" is overly broad. I suggest revising it to "Association of proteins with clinical outcomes in survival analyses in clinical trials" or a similar, more specific title that accurately reflects the content.

4. Figure 4: If the authors have not already, it would be valuable to address the difference in protein effect sizes between DECLARE-TIMI58 and EXSCEL trials in the Discussion. The stronger effects observed in EXSCEL warrant explanation or speculation on potential underlying factors.

5. Figures 5B and 5C: To improve visual interpretation, please consider adding color-coded squares surrounding the gene names based on their associated outcomes:

- HHF: Gray

- MACE: Blue

- RENAL: Orange

Keeping the gene names in black will ensure they remain easily readable.

I hope these suggestions are helpful to enhance the significance of the manuscript.

(Remarks on code availability)

Reviewer #2

(Remarks to the Author)

My comments have been thoroughly addressed in the revised version of this manuscript. The revised manuscript is also much easier to read and I think it has the potential to make an important contribution to the field.

(Remarks on code availability)

I quickly checked whether all analysis codes are provided on github, and this appears to be the case.

Reviewer #3

(Remarks to the Author)

I am happy with the revision that has addressed all the issues raised.

(Remarks on code availability)

Reviewer #4

(Remarks to the Author)

(Remarks on code availability)

Reviewer #5

(Remarks to the Author)

(Remarks on code availability)

Dear Reviewers,

Thank you for your valuable feedback. Our responses to each point are given below. In the manuscript, new text is blue, and the deleted text is captured via the track changes.

REVIEWER COMMENTS

Reviewer #1 (Remarks to the Author):

The study by Douglas P. Loesch et al. provides valuable insights into the complex pathophysiology of type 2 diabetes (T2D) and its associated comorbidities using a powerful proteogenomic approach. The authors leveraged the UK Biobank, a large population-based cohort, and two randomized controlled trials (EXSCEL and DECLARE-TIMI 58) to identify circulating protein biomarkers associated with polygenic risk scores (PGS) for T2D and related cardiometabolic conditions. The study's strengths include the large sample size, the use of multi-ancestry participants, and the validation of findings in independent clinical trial cohorts.

The authors identified a large number of proteins (839 for T2D PGS and 1,005 for cardiometabolic PGS) associated with genetic risk for T2D and its comorbidities. Some of these proteins, such as TFF3, EFEMP1, and MMP12, were further associated with renal and cardiovascular outcomes in the clinical trials, suggesting their potential as prognostic biomarkers. The study revealed shared pathways underlying T2D and its comorbidities, including the complement cascade, cholesterol metabolism, and IGF signaling.

Major comments:

The authors tackle an important topic and make use of valuable resources, especially those from EXSCEL and DECLARE-TIMI 58 trials, which are not publicly available. I appreciate the extensive analyses the authors have performed and their effort to make the data accessible through their web portal. To further enhance the clarity, coherence, and impact of the work, I would like to offer the following suggestions.

1. Clarifying the main purpose and key findings: I suggest the authors refine the narrative, text, and figures to more clearly convey the study's main objectives and the most important results. This will help readers better appreciate the significance of the work.

Thank you for this comment. We have taken several steps to address this.

Specifically, we:

- 1. added a flowchart to the main text (Figure 1) to help readers understand our study objectives and logic.**
- 2. added text in the introduction, discussion, and at key places in the results section to further clarify the logic (e.g. lines 88-96, 98-111).**
- 3. removed analyses that did not fit as readily with the rest of the manuscript or the did not yield meaningful results. Specifically, we removed NAFLD-related**

analyses, and analyses pertaining to diabetic complications (e.g., retinopathy). This helped focus the paper on T2D etiology along with the etiology of cardiorenal comorbidities.

2. Clarifying the connection between UKB and clinical trial analyses: To improve the clarity of the link between the analyses in UKB and the two clinical trials, I would recommend the authors elaborate on the logic, e.g., "We conducted extensive polygenic score (PGS), Mendelian randomization (MR), and mediation analyses in UKB to identify highly probable potential biomarkers. We then tested the associations between these biomarkers and cardiovascular events in the two clinical trials, replicating the N[number] protein-trait associations out of M[number] (FDR < XXX)."

Thank you for this comment. We have added language to the Introduction, Results, and Discussion to clarify the logic connecting the UKB and the clinical trials (e.g., lines 92-96 in the Introduction, lines 108-110 and 320-323 in the Results). Just to clarify, the aetiology of T2D and its comorbidities is a major focus of our manuscript as well as the identification of biomarkers. Our work identifies potential therapeutic targets as well as biomarkers that can provide insight to T2D progression.

3. Aligning the outcome traits between the analyses: Considering that EXSCCEL and DECLARE-TIMI 58 were primarily focused on cardiovascular outcomes in individuals with T2D, I would suggest the authors focus on cardiovascular events as the outcomes in the analyses in UKB rather than T2D itself or BMI. In addition, blood or plasma may not be the primary causal tissue for T2D and BMI.

Thank you for the comment. Our principal interest is in T2D aetiology and how it relates to comorbidities. The clinical trials were an excellent way to test for a link between the proteins we identified, including the T2D-susceptibility proteins, and the key comorbid outcomes captured in the clinical trials. We have further clarified this in the Introduction and the Discussion. However, we have removed analyses pertaining to NAFLD in the UKB as there was not an applicable liver-related endpoint in either clinical trial. We elected to keep renal outcomes as both trials contained a renal endpoint (in particular, DECLARE was designed with renal endpoints as an important secondary outcome). Finally, we agree that blood or plasma may not be the primary causal tissue for T2D and BMI; however, proteomics and outcomes data do not exist at scale for other potentially relevant tissues (e.g. adipose) as it is prohibitively expensive and technically complex to generate. We now discuss this as a limitation in the manuscript.

4. Highlighting important findings and their logic: Figures 3 and 4, which show the consistent association of IGFBPs with renal outcomes and major adverse cardiovascular events (MACE), appear to be crucial findings. I suggest the authors more clearly present the logic with which they narrowed down to this pathway through extensive analyses in UKB and clinical trials. If some analyses are not relevant to the main findings, they could be less emphasized or even omitted from the main texts.

We thank the reviewer for pointing this out. To understand if the PGS-associated proteins that we identified operate through some common biological pathways, we performed pathway enrichment on them and found several pathways that were enriched in multiple sets of PGS-protein associations. The IGF regulation by IGFBPs pathway was significantly enriched among multiple PGS-associated proteins, and as the reviewer mentioned, appeared in numerous analyses. As such, we have chosen to highlight it. We have clarified that further in the text (lines 386-389).

5. Streamlining supplementary materials: Given the extensive number (more than 60) of supplementary figures and tables, I suggest the authors select the important ones (less than 30 in total, for example) and simplify the overall logic of the study. Providing a table or a diagram summarizing which protein-trait relations are supported by each analysis could be helpful.

Thank you for the suggestion. We have now removed a number of supplementary figures, tables, and entire analyses to make the entire paper simpler to digest. This reduces the number of supplementary tables and figure to 44. As suggested, we have also added a table of major findings with our study-generated evidence, plus external evidence regarding druggability and tissue expression (Supplementary Table 21). We also hope that the online data portal accompanying this manuscript will provide a user-friendly way of browsing through the results / important findings from our analyses.

6. Improving transparency: I recommend clearly stating the thresholds of significance of each analysis in the main text.

Thank you for point this out, we have added significant thresholds throughout the text.

Minor comments:

- Line 45: "liver lipid" is duplicated.
- Line 54: "Some PGS-associated proteins ...". I suggest specifying the number of PGS-associated proteins that were associated with renal and cardiovascular trial outcomes.
- Line 359: To maintain consistency in target phenotypes and improve the rationale of the paper, I recommend considering the removal of diabetic retinopathy and hypertension from the causal inference using MR.
- Lines 368-369: I suggest clarifying whether the analysis using all diabetes-related ICD-10 codes (E10-E14) provided any additional findings compared to the analysis using only T2D (E11). If not, please consider removing the E10-E14 analysis, as type 1 diabetes (E10) and T2D (E11) are different diseases.
- Figure 4 B and D: The connection between plots and the protein names are vague. Please consider drawing lines between plots and protein names.

Thank you for these comments. We have now addressed these concerns. We removed diabetic retinopathy, hypertension, etc. We also removed analyses with non-T2D diabetes ICD-10 codes as they did not add anything substantial to the paper. We have also redrawn the figure to hopefully improve how the labels are connected to the points.

Reviewer #2 (Remarks to the Author):

This study investigated polygenic scores (PGS) for type 2 diabetes (T2D), for five clusters of T2D loci indicating genetic heterogeneity in T2D, and for selected T2D comorbidities – i.e., coronary artery disease (CAD), chronic kidney disease (CKD), body mass index (BMI), and non-alcoholic fatty liver disease (NAFLD) – for associations with close to 3,000 proteins in over 40,000 men and women in UK Biobank. The authors then tested the PGS-associated proteins for associations with incident cardiometabolic complications – i.e., insulin initiation, major cardiovascular events, hospitalization for heart failure, and renal complications – in two cardiovascular outcome trials among patients with T2D in a prospective analysis of data from two randomized controlled trials.

I think this study addressed a very relevant and important research question using an elaborate study design and applied solid statistical approaches. The work has the potential to contribute substantially to the field and can also inspire related field. The manuscript is largely well written; however, it was sometimes difficult to follow. Due to the complex design, it was not always easy to follow which exposure was linked to which outcome. I have several suggestions that may help to improve the accessibility of the manuscript. I also have several comments regarding the methods, which may need clarification.

1) Abstract: It would be helpful to list the T2D associated comorbidities that were studied in the abstract. Indicate in the abstract whether the proteins associated with T2D PGS were significant after multiple testing correction.

Thank you for this comment. Unfortunately, the formatting guide for the journal state that the abstract should be a maximum of 150 words. This made it challenging to add additional information to the abstract.

2) First paragraph of the introduction: Can the author give one example of how genome-wide association studies (GWAS) have identified genetic associations that have advanced our understanding of pathophysiological pathways underlying T2D? In the same sentence, can authors elaborate very briefly why connecting individual T2D risk variants to specific pathways remains challenging and how their study may address this challenge.

We have now added an example as suggested (C2CD4A in the first paragraph of the Introduction) and elaborated on the difficulties mapping T2D risk variants to specific pathways is challenging (lines 54-56). Note that the 2nd and 3rd paragraphs

of the Introduction builds upon this and discusses how this study potentially addresses the challenge of using genetic information to identify disease mechanisms.

3) Of the overlapping proteins measured across the two platforms (Olink and SomaScan), can the authors briefly elaborate on the comparability of measurements after normalization (e.g., in terms of abundance levels).

The excellent work by Eldjarn et al. in *Nature* 2023 (<https://doi.org/10.1038/s41586-023-06563-x>) performed this comparison in-depth, thus we have now mentioned this work in the cohort description section of Methods (lines 562 and 563). In short, Eldjarn et al found that abundance, and the results from pQTL analyses and protein-trait analyses, were moderately correlated. Differences in platforms likely results in higher false negative rate, but we would argue that results that are consistent across both platforms (e.g. consistent directions of effect) are likely robust.

4) Please clarify in the last paragraph of the UK Biobank phenotype definitions section whether the generation of summary statistics for the Mendelian randomization analysis are already published GWAS results, or it was performed by the authors. If they are published, more information from where they were extracted is needed.

We thank the reviewer for noticing this. Information regarding MR summary statistics was previously included in the Supplementary Methods. We have now moved this to the main Methods. For our Mendelian randomization analyses, we performed the GWAS ourselves within the UK Biobank, excluding participants from the proteomics study to allow for two-sample MR.

5) Provide abbreviations when first mentioned in manuscript (e.g. hospitalization for heart failure); also provide abbreviations in footnotes of all tables and figures.

Thank you for pointing this out, we have now addressed abbreviations in the manuscript.

6) It would be appreciated to present the correlations between the different PRSs on T2D, CAD, BMI, CKD, and the partitioned T2D PS. Also, between the genome-wide and GWAS-significant models.

We have now added a correlation heatmap of the different PGS used in this study to the Supplementary Figures (Supplementary Figure 7) and mention this result in the main text.

7) I was wondering what the rationale was of choosing PRS-CS/PRS-CSx to derive the

PRSs. Did the authors compare the performance with other software such as LDpred2-auto which showed to overcome PRS-CS?

This is a good question, thank you for bringing this up. LDpred2-auto does indeed outperform PRS-CS-auto for disease prediction, but we wanted to use a method that explicitly modelled the multi-ancestry GWAS data at which the T2D community has focused on. As such, we elected to use PRS-CSx as our primary PRS software. In the case of traits where we were unable to find multi-ancestry GWAS data (that were independent of the UK Biobank), we used PRS-CS to be methodologically consistent. PRS-CS/PRS-CSx also have an added benefit in that they require substantially less pre-processing compared to LDpred2 as they are more robust to violations to model assumptions. Also, while LDpred2 results in a score that outperforms PRS-CS, this improvement is relatively modest. This was consistent with our own preliminary observations; when we were first developing this study, we compared different PRS with varying predictive power and found that the number of significant protein associations did not differ substantially when the scores only moderately differed in their predictive power.

8) Where potential non-linear associations between PGS and proteins investigated – please clarify and if not, why not. Please clarify whether phenotypical or PGS BMI was used for adjustment in PGS-protein associations.

Thank you for this comment, we did not consider non-linear effects between the PGS and proteins. There has been very little evidence of substantial non-linear relationships between genetic variation and human traits in the literature. Such non-linear effects are hard to detect (given the very large sample sizes necessary for sufficient power) and, when detected, are often small in magnitude compared to linear effects. As such, we view the assessment of non-linear effects of PGS on protein levels as beyond the scope of this paper; however, it would be suitable for exploration in a separate study explicitly dedicated to finding said effects. For the BMI comment, we have only adjusted for phenotypical BMI (measured at baseline at the same time as blood collection) and have clarified that in the text.

9) I found it a bit confusing how you predicted the ancestries groups (end of PGS and protein associations in UK Biobank section). I suggest giving more detail on it. Was it based on the genetic principal components?

We thank the reviewer for this suggestion. More information regarding this was in the Supplemental Methods and has now been moved to the main Methods. We have also added more details in the corresponding section within the results. Yes, genetic prediction was based on principal components using the 1000 Genomes Project as a reference and the KING software (<https://www.kingrelatedness.com/ancestry/>).

10) It is confusing what the authors mean by this sentence: “We then compared the effect sizes (beta coefficients) of the PGS on circulating protein levels between ancestry groups using Pearson’s r and the slope of the regression line fitted to the beta

coefficients.” A clarification would be informative for the reader. Same is applicable for the section where these results are presented (page 15). Also, what was the proportion of different ancestries in the population?

Thank you for pointing this out, we have worked on those sections to improve their clarity. The UK Biobank, and the UKB-PPP proteomics subset, are both ~94% European ancestry, which we have also added to the main text. In addition, Supplementary Table 1 provides a more granular summary of predicted ancestries within each cohort.

11) In describing the mediation analysis, please clearly describe the exposure, the mediator, and the outcome. Also, at which level the exposure and mediator were modelled. Were proteins modelled in separate models or mutually adjusted for? How were mediator-mediator confounding and exposure-mediator confounding treated? A DAG would be helpful.

Thank you for bringing this to our attention, we have now more clearly stated the exposure, mediator, and outcome in the text. In our mediation analyses, the PGS was the exposure, the mediator was a circulating protein, and the outcome was an incident disease diagnosis (T2D, CAD, or CKD). The proteins were all modelled separately, and we did not use any multilevel or hierarchical models. Since the PGS is genetic and fixed at birth, much like in MR, there should not be any exposure-mediator confounding. However, as in MR analyses of complex traits and other similar published mediation analyses (e.g. Ritchie et al, *Nature Metabolism* 2021), mediator-mediator confounding is a real concern here due to pleiotropy. We had included this as a limitation in the text. We have now conducted sensitivity analyses which confirm that these models are susceptible to mediator-mediator confounding, and we have added this to the text as well.

12) In the 2-sample Mendelian randomization (MR) section further details are needed. It is not clear from where the data was extracted nor how many participants included the GWAS used in the analysis. How many genetic variants were included to build the genetic instruments? Was any filter applied to strengthen the genetic instrument (e.g., LD, R2...)? Also, are both samples (exposure and outcome) independent and from same population ancestries (these should be clarified to have insights of potential violation of the second MR assumption). I strongly suggest following the MR-STROBE available here: <https://www.strobe-mr.org/>. Additionally, please add the references for the several MR methods used.

We thank the reviewer for the clarifying questions and suggestions. The information regarding genetic instruments was previously in the Supplementary Methods, which we now moved to the main Methods. The genetic instruments were all obtained from the pQTL GWAS performed by Sun et al. (*Nature*, 2023) using the UKB-PPP proteomics data. They were identified as statistically independent by the authors and can be found in Supplementary Table 16 of Sun et al. We further filtered instruments by the F statistic and removed instruments if they were a pQTL

for 5 or more proteins. For the outcome, we performed the GWAS in the UK Biobank, excluding the UKB-PPP subjects, to ensure they were independent samples. All subjects were European ancestry in our primary MR analysis. We also ran a “multi-ancestry” MR, though just as a sensitivity analysis, which found very similar MR estimates. We have now expanded the MR methods section with this information as requested.

13) Cox PH regression: Please indicate the time scale of analysis; list all covariates either in the main text or if word count is an issue, in the footnotes of relevant tables and figures; where PH assumptions tested and satisfied? I think interactions between trial arm and proteins should be accounted for.

We have now created a new section in the main text describing the study populations which includes the time scale for the two clinical trials. The covariates are fully listed to the Methods section. For the interactions between the trial arm and the proteins, we agree that this could be a concern. In fact, the authors have discussed this before, and we have decided to redo the analysis using the placebo arm only. While this lowers the power of the analysis, the top results are still similar, and it avoids potential treatment/protein interactions. Finally, we tested all models for the PH assumption using the `cox.zph()` function from the survival package in R. If a model did not satisfy the assumptions, it was excluded from the corresponding results section, and the results of the `cox.zph()` function are now included in the survival results. Overall, with these changes, our results are similar and thus conclusions unchanged.

14) Briefly describe the different study populations.

Thank you for suggesting this, the first subsection of the Results section now describes the study populations.

15) Page 13, first two sentences of results can rather be moved to into introduction or discussion; an understanding of the study population would be more informative.

Thank you, we have removed those sentences and as stated above, we have now added a study population section.

16) When reporting results, clarify whether reported associations were observed after multiple testing correction.

Thank you for the suggestion, we have now clarified whether the reported association was after a multiple testing correction. Essentially, multiple-testing correction was applied to all results.

17) I find it confusing to say that protein effect sizes were correlated – effect size is not a unit of a variable. I guess that beta coefficients were extracted and correlated? Please clarify what was done, which correlation method was used, and modify throughout the

manuscript.

That is correct, it was indeed beta coefficients and Pearson's correlation was used. We have modified the manuscript accordingly.

18) Please add the estimate and confidence interval in the results for ERBB4 in page 16.

Thank you for pointing this out, we have now added MR estimates and confidence intervals for ERBB4.

19) First paragraph of discussion could rather go to the second paragraph of discussion with relevant references.

Thank you for this suggestion, we have re-worked the Discussion section according to suggestions of another reviewer which also addressed this suggestion.

20) It is not always clear to the reader when authors talk about mediation results in terms of exposure, mediator, and outcome. E.g., second paragraph of the discussion, FAM3D mediated which relationship?

We have further clarified the relationship between exposure, mediator, and outcome in the text. In this section, FAM3D mediated the beta cell score's effect on incident T2D. We have clarified that in the text as well (lines 422 and 423).

21) First sentence, fourth paragraph: it is unclear what is meant with "inverse relationship between T2D and CKD risk" – were associations between T2D and CKD risk investigated?

We have further clarified this in the text (lines 490-492). What we were referring to is that relatively lower levels of IGFBP2 were associated with risk of incident T2D while relatively higher levels of the IGFBP2 were associated with incident kidney disease, as has previously been reported in the literature. This is something we feel warrants further investigation in subsequent works.

22) Figure 3, Since HR at months 12 are almost consistently larger, can this indicate reverse causation? Did the authors consider cumulative exposure to proteins, e.g., by adjusting for baseline. What would complete concordance tell us?

Thank you for this suggestion. It is true that the hazard ratios are larger at the second timepoint. While this could be due to reverse causation, we would also expect that for a truly causal protein the higher HR could be more cumulative exposure from the protein. We do note that the higher HRs at the second timepoint are most pronounced for the renal outcomes in both trials. In this case, there certainly could be reverse causation, since as the kidneys fail it would be expected that there will be some dramatic increases in levels of some circulating proteins. We have now adjusted for baseline measurements, which did reduce this trend for the renal outcomes. However, for several proteins (particularly for cardiovascular

outcomes), their HR was still larger in the second timepoint, and for some, the HR increased with the baseline adjustment. It is also worth noting that protein biomarkers do include a temporal element to them, so it is possible they are better predictors of an event when the measurement occurs closer in time to the event.

In summary, we do find it challenging to distinguish reverse causation from the combination of forward causation with feedback mechanisms or cumulative exposure. However, this does not impact our core conclusions as the trials simply add value for linking our already PGS-protein associations with clinical trial outcomes.

23) Figure 4, Kaplan-Meier is little informative - one cannot distinguish the two groups.

Thank you, that is a fair point. The sample size of DECLARE certainly limits the how informative the Kaplan-Meier can be. We removed the Kaplan-Meier panels from the figure.

24) Use person first language throughout the manuscript including supplementary material: e.g., patients with T2D or adults with T2D instead of “diabetics” or “T2D patients”.

Thank you for pointing this out, we have indeed missed some instances where we did not use person first language, particularly in the supplement. We have now addressed this.

Reviewer #3 (Remarks to the Author):

The paper by Loesch DP E et al examined the causal associations of circulating proteins with polygenic scores for T2D and its main cardiometabolic complications, utilising newly emerging Olink proteomics data in the UK Biobank, with further follow-up replications and explorations of findings in two randomised trials. T2D is a common but highly heterogeneous conditions. Although a large number of genetic variants have been identified, the biological mechanisms underlying T2D genetic risks and the molecular characteristics of different genetic subtypes of T2D defined by polygenic scores (e.g. beta cell, lipodystrophy, obesity, liver lipid, and proinsulin) remain poorly understood. The present study represented the largest investigation to date on plasma protein markers underlying genetic risk of T2D and the related complications, using a range of analytic methods (e.g. MR, mediation, pathway enrichment analyses). The study designs are appropriate, analytic methods and procedures robust, and study limitations well recognised. Importantly, the study has generated many informative and novel findings that have the potential to improve understanding of T2D aetiology and inform risk prediction, patient stratification and development of new treatments. However, there are several general and specific issues related to data analyses, presentation and interpretation, which could be further clarified and improved.

1. In various analyses, BMI was adjusted. While appropriate, it would be helpful to also consider adjusting for other measures of adiposity, particularly WC and WHR. As measures of central obesity, WC and WHR have been shown in both conventional and genetic analyses to be more strongly associated with risk of T2D compared with BMI. Moreover, there is emerging evidence from recent proteomics-related research that the disease-causing pathways associated with central obesity differ somewhat from those associated with BMI. Hence, it would be helpful to look into this separately and compared the findings with those related to BMI.

Thank you for this suggestion, as requested we have now included an analysis where we adjusted T2D PGS-protein associations for baseline BMI, WC, or WHR. Comparisons of the results showed that they were broadly similar, so we left our main results as is in the text. However, we do provide all the results as a supplementary table (Supplementary Table 7) to look up individual proteins that might differ based on the adiposity measurement used.

2. Examine and clarify the effects of potential collider/selection bias in the following scenarios:

a. Adjusting for BMI when assessing the association of PGS with proteins, in which BMI can be considered as a mediator between PGS and proteins, leading to potential collider bias.

b. Applying BMI-adjusted GWAS for the MR analysis would be subject to collider bias. MVMR would be an alternative method, although instrument used may be weaker (the conditional F statistics could be checked beforehand).

c. When assessing the association of PGS with proteins in the two RCT patient populations (EXSCEL and DECLARE-TIMI 58), T2D is likely a mediator between PGS associated proteins and clinical trait outcomes, although the 3 models might address the confounders.

It would be helpful to undertake additional analyses and to discuss their likely impacts and acknowledge these as potential study limitations.

Thank you for these suggestions. Regarding part “a” of this comment, we have added a comment regarding collider bias in the discussion (Lines 507-512). Also, we have now performed mediation analysis with T2D PGS as the exposure, BMI as the mediator, and the protein as the outcome, results of which are on lines 135-145. Briefly, for 111 proteins, the PGS’s direct effect was mostly mediated by BMI (e.g., Leptin). For the remaining proteins associated with the genome-wide T2D PGS, the direct effect of the PGS was statistically significant, but the proportion of total effect mediated by BMI can range from 0.07 to 0.76, with a median of 0.41. This better characterizes the relationship between the T2D PGS, BMI, and the circulating proteins. For part b of the comment, we have done as suggested and added in an analysis using MVMR for the loci with appropriate instruments for both the BMI and protein exposures. Finally, all participants in the trials have T2D, so we recognize that collider bias can still occur in the case of proteins/pathways that lead to the development of T2D. We added that as a limitation in the Discussion section.

3. It is not very clear as to why the median of all the MR estimates (IVW, weighted IVW, median, and Egger methods, as well as the p-value and 95% CI) was used to interpret the MR results. Each method has its assumptions for specific purpose. Simplifying all the estimates into one median estimate will likely lose information. For example, an “outlier” MR estimate may provide specific insights in some cases. Besides, the median estimate would be driven by how many MR methods that have been applied. It would be helpful to provide further details in interpreting the MR findings based on ensemble MR estimate.

Thank you for this comment, we agree that combining estimates into a single “consensus” estimate loses information. While we still use the median p-value as a filtering method, we now report all MR estimates separately, including in the figures. We maintain median p-values as an ensemble MR estimate to make the results for readable overall and as a simple initial screen, but hopefully in presenting all MR estimates our readers may instead assess their assumptions on a case-by-case basis, depending on their protein/pathway of interest. In the MR results we discuss in the text, the MR estimate is consistent across methods, though the MR-Egger standard errors tend to be larger. For T2D, NUCB2 was only protein where an MR method produced a result inconsistent with the other methods. In this case, while the IVW, simple median, weighted median, and MR-Link-2 methods all produced a negative MR estimate, the Egger method resulted in a positive MR estimate with a very wide 95% confidence interval.

4. Likewise, it is not clear as to why the MR analyses were confined only to those proteins with ≥ 3 cis-/trans-pQTLs. Although trans-pQTLs need to be handled with caution, there is currently no general consensus as to how the cis- and trans-pQTLs should be incorporated into a single MR estimate. Besides, some novel methods have recently been proposed (e.g. MR-Link-2, MR-BMA), which may provide a useful alternative to ensemble MR approach, which could be over-simplified. Moreover, to help establish causality reliably, it may be more appropriate to use the ensemble MR estimate only for screening purpose. But still, it is necessary to triangulate all the MR (including all pleiotropy robust estimates, cis-pQTLs-based MR estimates, and cis- and trans-pQTLs-based MR estimates) and the colocalization evidence on the candidate proteins obtained from the screening step.

Thank you for this suggestion. To address this comment, we have substantially overhauled the MR section of the manuscript, presenting the results from *cis* MR, *cis/trans* MR, and colocalization. As requested, we have also run MR-Link-2 and present the results from that analysis alongside the more convention MR analyses that we have already done. As stated above, we also only use the ensemble p-value as a screening step to identify potentially causal proteins. Overall, our results are consistent with previous observations and our main conclusions are unchanged, though we now discuss additional proteins in the text, particularly those identified via *cis/trans* MR.

5. While appropriate and informative, the reverse MR seemed to have involved all rather

than only PGS-associated proteins. As such the study findings may be subject to reverse causality, for the proteins caused by specific diseases should not be considered as the outcomes in this particular setting.

While we appreciate the reviewer's comment, we are unclear how our current approach makes the analysis more susceptible to reverse causality. In performing reverse MR for all proteins, one would think that we are being more conservative/comprehensive in identifying proteins for which the directionality of the association was ambiguous.

6. Given the range of analyses undertaken and study findings generated, it would be helpful to provide a summary table for all or selected top causal proteins identified, showing all the key findings, including tissue expression, secreted or not, evidence of drug development, overlap across different disease outcomes, etc. This should help readers to fully understand and appreciate the breadth and depth of the findings and opportunities for further translational research.

This is a great suggestion which certainly increasing the paper's clarity and impact; we have now generated a table (Table 21) with both our evidence alongside external information regarding druggability, drug modality, tissue expression, and so forth.

7. While all the key elements are properly described and discussed, the writing could be further restructured and improved. For example, in the Results section, there are many undue repetitive justifications for specific analyses and use of certain datasets (e.g. lines 3-6 in Page 13; lines 15-18 in Page 17; lines 14-18 in Page 19), which are properly described in the Methods section. Similarly, there are many undue discussion of the study findings in the Results section (e.g. last three lines in Page 13), which makes the Discussion section rather weak and lightweight. This is particularly the case for the Results section related to two randomised trials, which could be shorten significantly without losing any key messages. In discussion section, it would be helpful to highlight briefly the study strengths in the context of existing knowledge and research methods, prior to discussion of its limitations.

Thank you for this suggestion, we have re-worked the Results section to hopefully improve its impact and went through the highlighted sections to remove repetition and misplaced discussion. However, we have maintained introductions to the Results subsections. While they are described in the Methods section, we do feel that it improves readability since this journal uses a "methods last" format.

Reviewer #4 (Remarks to the Author):

Reviewer #5 (Remarks to the Author):

Reviewer #5 (Remarks on code availability):

NA

Dear Reviewers,

Thank you for your valuable feedback. Our responses to each point are given below. In the manuscript, changes to the text are indicated in red.

REVIEWERS' COMMENTS

Reviewer #1 (Remarks to the Author):

I thank the authors for addressing comments from me and other referees. The manuscript has been significantly improved since the previous submission. To help further refine the work, I've outlined some suggestions below:

1. Abstract: The abstract could be further refined to better reflect the study's key findings and methodologies. The introductory sentences (lines 38-43) may be a bit wordy and can be condensed. Please consider describing the downstream analyses such as causal inference, pathway enrichment, and Cox regression more in detail.

Thank you for this suggestion. We have trimmed the content in lines 38-43, remove seven words. Even after condensing those lines, the format guide states that the abstract should be 150 words at length, which prevents providing more details. However, we can ask if there is any flexibility here, and if there is, add these details.

Minor comments:

2. Figure 2C: please consider changing "T2D_PGS" to "PGS_T2D_gw" for consistency and clarity.
Thank you for noticing this, we have changed this in Figure 2C.

3. Figure 4: The current title "Survival analysis in the clinical trials" is overly broad. I suggest revising it to "Association of proteins with clinical outcomes in survival analyses in clinical trials" or a similar, more specific title that accurately reflects the content.

Thank for this suggestion, we have changed the title to "Association of proteins with clinical trial outcomes and survival analyses."

4. Figure 4: If the authors have not already, it would be valuable to address the difference in protein effect sizes between DECLARE-TIMI58 and EXSCEL trials in the Discussion. The stronger effects observed in EXSCEL warrant explanation or speculation on potential underlying factors.

We agree, this needs to be addressed in the Discussion. There are potentially two things at play here. First, EXSCEL is a pragmatic trial and as such has relaxed entry requirements, and second, the renal outcomes do differ between the two clinical trials. For EXSCEL, the renal endpoint is two consecutive measurements of eGFR < 30 ml/min/1.73m², while for DECLARE-TIMI 58, the renal endpoint is a

composite renal outcome comprising of a sustained decrease of 40% or more in eGFR to < 60 ml/min/1.73m², new end-stage renal disease, or death from renal causes. We have now added this information to the Discussion.

5. Figures 5B and 5C: To improve visual interpretation, please consider adding color-coded squares surrounding the gene names based on their associated outcomes:

- HHF: Gray
- MACE: Blue
- RENAL: Orange

Keeping the gene names in black will ensure they remain easily readable.

I hope these suggestions are helpful to enhance the significance of the manuscript.

Thank you for this suggestion. The R package we used for plotting the labels, ggrepel, treats the text and the border as one object and it is not possible to set the color of the border without changing the color of the text. We did change the color of the line segment connecting the point to the label, and we will also communicate with the journal to see if this can be manually changed by the publication team.

Reviewer #2 (Remarks to the Author):

My comments have been thoroughly addressed in the revised version of this manuscript. The revised manuscript is also much easier to read and I think it has the potential to make an important contribution to the field.

Thank you for your feedback.

Reviewer #2 (Remarks on code availability):

I quickly checked whether all analysis codes are provided on github, and this appears to be the case.

Reviewer #3 (Remarks to the Author):

I am happy with the revision that has addressed all the issues raised.

Thank you for your feedback.

Reviewer #4 (Remarks to the Author):

Reviewer #5 (Remarks to the Author):
